# An extended time-series (2000-2018) of global NPP-VIIRS-like nighttime light data from a cross-sensor calibration

Zuoqi Chen[1,2], Bailang Yu[3,4], Chengshu Yang[3,4], Yuyu Zhou[5], Xingjian Qian[3,4], Congxiao Wang[3,4], Bin Wu[3,4], Jianping Wu[3,4]

[1]Key Laboratory of Spatial Data Mining and Information Sharing of Ministry of Education, National & Local Joint Engineering Research Center of Satellite Geospatial Information Technology, Fuzhou University, Fuzhou 35002, China
[2]The Academy of Digital China, Fuzhou University, Fuzhou 350002, China
[3]Key Laboratory of Geographic Information Science (Ministry of Education), East China Normal University, Shanghai 200241, China

[4]School of Geographic Sciences, East China Normal University, Shanghai 200241, China
[5]Department of Geological and Atmospheric Sciences, Iowa State University, Ames, IA 50011, USA

*Correspondence to:* Bailang Yu (blyu@geo.ecnu.edu.cn)

**Abstract:** The nighttime light (NTL) satellite data have been widely used to investigate urbanization process. The Defense Meteorological Satellite Program-Operational Linescan System (DMSP-OLS) stable nighttime light data and Suomi National

Polar-Orbiting Partnership-Visible Infrared Imaging Radiometer Suite (NPP-VIIRS) nighttime light data are two widely used NTL datasets. However, the difference of their spatial resolutions and sensor design makes these two datasets can not be directly used together for a long-term analysis of urbanization. To solve this issue, an extended time-series (2000-2018) of NPP-VIIRS-like NTL data were proposed in this study through a cross-sensor calibration from DMSP-OLS NTL data (2000-2012) and a composition of monthly NPP-VIIRS NTL data (2013-2018). Compared with the annual composited NPP-VIIRS

NTL data in 2012, our product of extended NPP-VIIRS-like NTL data shows a good consistency at the pixel and city levels with $R^2$ of 0.87 and 0.95, respectively. We also found that our product has a good accuracy by comparing with DMSP-OLS radiance calibrated NTL (RNTL) data in 2000, 2004, 2006, and 2010. Generally, our extended NPP-VIIRS-like NTL data (2000-2018) have a good spatial pattern and temporal consistency, which are similar to the composited NPP-VIIRS NTL data. In addition, the resulting product could be easily updated and provide a useful proxy to monitor the dynamics of demographic

and socio-economic activities for a longer time period compared to existing products. The extended time-series (2000-2018) of nighttime light data are freely accessible at https://doi.org/10.7910/DVN/YGIVCD (Chen et al., 2020).

## 1. Introduction

Along with the artificial electric light has been widely equipped in most buildings and infrastructures, the nighttime light (NTL)

remote sensing data have been widely used to investigate human activities (Gaston et al., 2013;Falchi et al., 2011;Elvidge et al., 1997a;Baugh et al., 2013;Li et al., 2018). So far, two NTL data, the Defense Meteorological Satellite Program-Operational

Linescan System (DMSP-OLS) stable nighttime light data and Suomi National Polar-Orbiting Partnership-Visible Infrared

Imaging Radiometer Suite (NPP-VIIRS) nighttime light data, are widely used to monitor and diagnose urbanization process.

For instance, the population (Sutton et al., 2001;Xu et al., 2015;Elvidge et al., 1997b;Yu et al., 2018) and economic

development (Zhao et al., 2017;Lo, 2002;Ma et al., 2012;Yu et al., 2015;Zhao et al., 2019b) has been successfully estimated,

the energy consumption (Shi et al., 2018;Shi et al., 2016b) and environment issues (Ou et al., 2013;Shi et al., 2016a;Liu et al.,

2018;Jiang et al., 2018) have been well revealed, and the urban area (Shi et al., 2014a;Cao et al., 2009;Zhou et al., 2014;Chen

et al., 2019;Zhou et al., 2015) and its spatial structure (Chen et al., 2015;Lu et al., 2018;Chen et al., 2017b;Yu et al., 2014;Wu

et al., 2019) have also been effectively detected. Therefore, these two NTL datasets are both good proxies of detecting the

dynamics of demographic and socio-economic activities at different spatial scales (Yang et al., 2019).

Unfortunately, these applications were always limited by the two NTL datasets' quality and available time span. The DMSP-

OLS NTL annually composited data are available from 1992 but ended in 2013 (Fig. 1). It has three issues, including lack of

on-orbit radiance calibration, saturation issue and blooming issue (Letu et al., 2010;Cao et al., 2019;Elvidge et al., 2014;Levin

et al., 2020), which limit its potential applications. The NPP-VIIRS NTL data have a better data quality (e.g., higher spatial

resolution of ~500 m, etc.) and a superior detection ability, but its short available time span limited its power for long-term

analysis and applications. As shown in Fig. 1, the monthly composited data is from April 2012 to present, while the annual

NPP-VIIRS NTL data covers only two years: 2015 and 2016. In addition, DMSP-OLS NTL data records the digital number

(DN), which is totally different from the radiance value in NPP-VIIRS NTL data. Consequently, these two NTL data are not

comparability and could not directly be used together. Therefore, an extended time-series of nighttime light data with an

appropriate quality and a better consistency becomes critical for the further temporal nighttime light applications.

(Insert Figure 1 near here)

Until now, literatures are limited for extending NTL data by integrating DMSP-OLS data and NPP-VIIRS NTL data (JESWANI,

2017). Among them, Shao et al. (2014), according to the NPP-VIIRS Day/Night band data and lunar irradiance model,

developed a vicarious radiometric calibration for DMSP-OLS daily NTL data. However, this model requires selecting specific

events at night as criteria and is not suitable for the annual DMSP-OLS NTL composite data. Zhu et al. (2017) and Li et al.

(2017) both attempted to use a power function for integrating the DMSP-OLS NTL data and NPP-VIIRS NTL data. Zhu et al.

(2017) fitted the power function by using the cumulated DMSP-OLS and NPP-VIIRS NTL intensity within each province in

China from 1992 to 2015. Then this power function was applied to the cumulated NPP-VIIRS NTL intensity to generate

simulated DMSP-OLS NTL intensity. But the power function from Li et al. (2017) was fitted from annual DMSP-OLS NTL

data and monthly NPP-VIIRS NTL data. This power function was then conducted to inter-calibrate these two NTL data for

analyzing Syria's major human settlement loss during a war. These power functions are both heavily relied on the strategy of

training samples selection and is not easy to be extended to other world regions or the entire world. Instead of using power



function and traditional DMSP-OLS stable NTL data, Zheng et al. (2019) conducted a geographically weighted regression model to fit the radiance-calibrated DMSP-OLS NTL data and NPP-VIIRS NTL data, then generated the DMSP-like NPP-VIIRS NTL data for the further research. Zhao et al. (2019a) proposed a sigmoid function model with a series of pre-processing procedures to convert NPP-VIIRS NTL data into simulated DMSP-OLS NTL data from 1992 to 2018 in Southeast Asia. Li et al. (2020) provided a global DMSP-OLS-like NTL data, called as harmonized DMSP-OLS NTL data, through a stepwise calibration of DMSP-OLS NTL data and an kernel density-based integration of calibrated DMSP-OLS and NPP-VIIRS NTL data. However, since NPP-VIIRS NTL data has a better quality than DMSP-OLS NTL data, the performance of estimating social-economic index and extracting urban spatial structure from NPP-VIIRS NTL is much higher (Shi et al., 2014b;Chen et al., 2017b). Therefore, simulating an extended time-series of NPP-VIIRS-like NTL dataset other than traditional DMSP-OLS-like NTL data, would be much helpful for the further analysis and applications. Given the difficulties mentioned above, a new approach to cross-sensor calibrating these two NTL data is still challengeable.

Recently, deep learning technologies present a great potential for image process, such as image restoration, image denoising, and target recognition/classification (Goodfellow et al., 2016). An auto-encoder model proposed by Hinton and Zemel (1994) contains a set of recognition weights for encoding the input data and a set of generative weights for reconstructing a similar input data. With convolutional neural networks (CNN), the auto-encoder model becomes more powerful for learning image high-level features and enhancing the input image quality (Wang and Tao, 2016;Jain and Seung, 2009). For instance, Tan and Eswaran (2008) and Vincent et al. (2010) successfully developed a stacked auto-encoder network with CNN to reconstruct and denoise handwritten digital images, respectively. Chen et al. (2017a) applied a Residual Encoder-Decoder Convolutional Neural Network (RED-CNN) to enhance a CT image from low-dense to normal dense. By analogy, the DMSP-OLS NTL data are like the low-dense image with some noises, while the NPP-VIIRS NTL data can be treated as the high-quality image. According to the successful cases from the literatures mentioned above, we deeply believed the auto-encoder model has an ability of converting DMSP-OLS NTL data to NPP-VIIRS NTL data.

In this study, we developed an auto-encoder (AE) model including convolutional neural networks to integrate DMSP-OLS NTL and NPP-VIIRS NTL data, and generated an extended time-series of global annual NPP-VIIRS-like NTL data from 2000 to 2018. The Google Earth Engine (GEE) platform (Kumar and Mutanga, 2018) and a parallel computing platform named as Compute Unified Device Architecture (CUDA) of the graphics processing unit (Huang et al., 2015) were used in the proposed framework. The remainder of this paper is organized as follows. Section 2 describes the data involved in this study and the illustration of data preprocessing. The auto-encoder network structure and cross-sensor calibration are presented in Section 3. In Section 4 and 5, the extended time-series of global NPP-VIIRS-like NTL data, accuracy evaluation, spatial pattern and temporal consistency are discussed. The findings are summarized in the final section.



## 2. Data

This study used three datasets (Table 1). The first one is the enhanced vegetation index-adjusted NTL index (EANTLI) of
2000-2013 (Zhuo et al., 2015), as an input data in the AE model, from the annual calibrated DMSP-OLS NTL data and
Enhanced Vegetation Index (EVI) products from Moderate Resolution Imaging Spectroradiometer. The second dataset is the
composited NPP-VIIRS NTL data from 2012 to 2018, which was annually summarized from the monthly NPP-VIIRS NTL
data. This 2012 composited NPP-VIIRS NTL data was used as the label data in AE model while 2013 being the reference for
validation, and 2013-2018 were appended to the NPP-VIIRS-like NTL data (2000-2012) simulated by AE model. The third
one is the DMSP-OLS radiance calibrated NTL (RNTL) data, which were for the validation procedure.

(Insert Table 1 near here)

### 2.1. Enhanced vegetation index-adjusted NTL index (EANTLI)

Due to the mentioned issues in the annual DMSP-OLS NTL data, an enhanced vegetation index-adjusted NTL index (EANTLI)
of 2000-2013 was used as an input in the AE model to simulate NPP-VIIRS-like NTL data. EANTLI was proposed by Zhuo
et al. (2015) by fusing the EVI and DMSP-OLS NTL data. The EANTLI not only reduces the saturation issues, but also
enhances the nighttime light intensity variations. It can be expressed mathematically in the equation:

$$\text{EANTLI} = \frac{1+(nNTL-EVI)}{1-(nNTL-EVI)} \times NTL \tag{1}$$

where $EVI$ represents the value from the annual average EVI value, $NTL$ is the DMSP-OLS NTL intensity, while $nNTL$
indicates the normalized $NTL$.

In this study, to get an EANTLI dataset with a better temporal consistency, the calibrated DMSP-OLS NTL data replaced the
original DMSP-OLS NTL data in Eq. (1). The calibrated DMSP–OLS NTL of 2000–2013 used in this study were derived from
the original annual DMSP-OLS NTL data via a stepwise calibration (Li and Zhou, 2017). The calibrated DMSP–OLS NTL
data have a better temporal consistency and a greater agreement with the NPP-VIIRS NTL data (Li et al., 2020). This calibrated
NTL data covers the same extent (180 to 180 degrees longitude and -65 to 75 degrees latitude) and spatial resolution (30 arc-
seconds). The data record the digital number (DN) values with a range from 0 to 63. 0 indicates the no nighttime light intensity,
as background, while 63 indicates the highest nighttime light intensity.

The 16-Day EVI products (MOD13A1) with spatial resolution of 500 m are involved. To mitigate the sensitivity to seasonal
and inter annual fluctuations, the MOD13A1 EVI products were processed as an annual average EVI (Jing et al., 2015). It is
worthy noted that since the MOD13A1 EVI data is only available from 2000 onwards, we only processed the Enhanced
vegetation index-adjusted NTL index (EANTLI) of 2000-2013. More details and advantages of calibrated DMSP–OLS NTL
and EANTLI can be found in Li et al. (2020) and Zhuo et al. (2015).



### 2.2. Composited NPP-VIIRS NTL data

The monthly NPP-VIIRS NTL data were calibrated and aggregated to the annual NPP-VIIRS NTL data from 2012 to 2018. It is worthy to note that the composited NPP-VIIRS NTL data of 2012 was for the validation process, the composited NPP-VIIRS NTL data of 2013 was for the training process in AE model, and the composited NPP-VIIRS NTL data from 2013 to 2018 were appended to the final product as a part of NPP-VIIRS-like NTL data.

The version 1 monthly NPP-VIIRS NTL composite data (vcm version) of April 2012 to December 2018 provided by Colorado School of Mines were used to composite annual NPP-VIIRS NTL data. The monthly NPP-VIIRS NTL data cover the same extent of calibrated DMSP-OLS NTL data with a finer spatial resolution of 15 arc-seconds (approximately 500 m near equator) and a more sensitive sensor with a unit of nano-Wcm$^{-2}$sr$^{-1}$. Since the official annual NPP-VIIRS NTL data are only available in 2015 and 2016 and it requires inaccessible parameters to repeat the official annual composite process for other years (Elvidge et al., 2017), we composited a new annual NPP-VIIRS NTL data by using the median value of 12 NPP-VIIRS NTL monthly composite data per each pixel. To differentiate with the official annual NPP-VIIRS NTL data (2015 and 2016), we named our annual NPP-VIIRS NTL data as composited NPP-VIIRS NTL data in the following sections.

We used the median value, instead of the traditional average or max value because the monthly NPP-VIIRS NTL data were contaminated by stray light in the mid-to-high latitudes region during the entire summer and the contaminated pixels are reassigned as 0 in the official monthly composite data (Elvidge et al., 2017). As the mean value could lower the NTL intensity and the max value could highlight abnormal value, the median value could be more reliable than the mean or max value (Liu et al., 2010). A validation of this median composited NPP-VIIRS NTL was conducted as described in the following section.

According to the correction model of NPP-VIIRS NTL data proposed by Shi et al. (2014b) and Ma et al. (2014), a dark background mask and a nighttime light intensity threshold value are required to remove pixels of unstable and abnormal nighttime light intensity, respectively. First, the dark background mask consists of the EANTLI pixels with value of 0 and the NPP-VIIRS NTL pixels with intensity lower than 1 nano-Wcm$^{-2}$sr$^{-1}$. We then filtered the corresponding annual NPP-VIIRS data by using this dark background mask. Then, according to global city rank from the Globalization and World Cities (GaWC) Research Network (Taylor et al., 2010), Shanghai and Beijing in China and New York City in USA were selected in this study to find out their highest nighttime light intensity as a threshold value for each year. Once an NPP-VIIRS NTL pixel value is higher than this threshold value, this pixel value will be replaced by its maximum NTL intensity within its eight neighbor pixels to eliminate the abnormal value.

### 2.3. DMSP-OLS radiance calibrated NTL (RNTL) data

DMSP-OLS RNTL data in 2000, 2004, 2006, and 2010 were selected as reference data to evaluate if our product has a good accuracy during the entire time series, since the NPP-VIIRS NTL data is only available after 2012. The DMSP-OLS RNTL



data have a radiance calibration based on pre-flight sensor parameters and are free from sensor saturation (Feng-Chi et al.,
2015). However, this dataset is still unitless, because the lacking of on-board calibration system for all DMSP-OLS makes the
degradation of sensors over time can not be measured precisely. DMSP-OLS RNTL data has the same extent of the original

DMSP-OLS NTL data with the same spatial resolution (30 arc- seconds), but it is only available in specific years. The four
selected DMSP-OLS RNTL data were accessed from Colorado School of Mines
(https://eogdata.mines.edu/dmsp/download_radcal.html).

## 3.    Methodology

As outlined in Fig. 2, after data preprocess mentioned above, a four-step approach was proposed to generate an extended time-

series (2000-2018) of NPP-VIIRS-like NTL data from the EANTLI data and composited NPP-VIIRS NTL data. In step 1, a
modified auto-encoder model was developed. In step 2 and 3, the architecture of auto-encoder model with CNN was designed
and the cross-sensor calibration model was trained. In step 4, an extended time-series (2000-2018) of NPP-VIIRS-like NTL
dataset was generated using the trained model by inputting the 2000-2012 EANTLI data and appending the composited NPP-
VIIRS NTL data (2013-2018). Finally, a comprehensive accuracy evaluation was conducted.

165                                              (Insert Figure 2 near here)

### 3.1.  The auto-encoder (AE) model

The auto-encoder model was trained by setting the 2012 EANTLI as input data and the 2012 composited NPP-VIIRS NTL
data as label. Then, the trained model was adopted to simulate the NPP-VIIRS-like NTL data by inputting the 2000-2011
EANTLI data. The auto-encoder model includes two main parts (encoder and decoder) as shown in Fig. 3. Let $\mathbf{X} \in \mathbf{R}^{m \times n}$,

$\mathbf{Y} \in \mathbf{R}^{m \times n}$, and $\hat{\mathbf{Y}} \in \mathbf{R}^{m \times n}$ be the annual EANTLI data, composited NPP-VIIRS NTL data, and simulated NPP-VIIRS-like
NTL data. The encoder part is to learn a deterministic mapping $f_\theta$ which could transfer $\mathbf{X}$ into a hidden representation ($\mathbf{H}$).
A typical deterministic mapping function can be expressed as:

$$f_\theta(x) = s(Wx + b) \tag{2}$$

where $x \in \mathbf{X}$, and $\theta$ represents the parameter set, including weight matrix ($W$) and offset ($b$). On the contrary, the traditional

decoder part is to reconstruct $\mathbf{X}$ using the high-level features extracted from the hidden representation ($\mathbf{H}$). This reconstruction
$g_{\theta'}$ is called as decoder and can be expressed as:

$$\hat{\mathbf{Y}} = g_{\theta'}(h) = s(W'h + b') \tag{3}$$

where $h \in \mathbf{H}$, and $\theta'$ represents the parameter set, including weight matrix ($W'$) and offset ($b'$). However, in this study the
decoder part was modified to map the composited NPP-VIIRS NTL data ($\mathbf{Y}$), rather than the traditional reconstruction of $\mathbf{X}$,



which means that the problem can be transformed to build two functions $f_\theta$ and $g_{\theta'}$ via deep learning technology to minimize a specific loss function (e.g., Mean Square Error):

$$arg \min_{f,g} \|\widehat{Y} - Y\|^2 \tag{4}$$

(Insert Figure 3 near here)

### 3.2. The Auto-encoder with a CNN architecture design

AE and CNN have both demonstrated an excellent performance on the image feature extractions. In this study, based on AE and CNN framework, we proposed a 10-layers network architecture as the cross-sensor calibration model (Fig. 4). This model includes five convolutional operations for encoding EANTLI data and then stacks 5 deconvolutional operations for decoding in symmetry. Because our intention is to reconstruct images, instead of to classify targets, the fully-connected layers in traditional encoder and decoder parts were removed from our architecture.

190                              (Insert Figure 4 near here)

The kernel size of convolutional and deconvolutional operations adopted in this architecture is 3 by 3 with stride and padding of 1 to maintain the size of output layers the same as its input layer. In the encoder part, the batch normalization (BN) operations were added after each convolution layer to keep from the vanishing or exploding gradient problem (Ioffe and Szegedy, 2015). The rectified linear unit (ReLU) function was applied in this architecture as the activation function after each convolutional

and deconvolutional layer, except the last deconvolutional layer. The ReLU function can be formed as:

$$\text{ReLU}(x) = \max(0, x) \tag{5}$$

The traditional CNN structure always contains more than one pooling operation to improve its learning efficiency, but these pooling operations could lose the details of the input images. To keep the image information as much as possible, the pooling processes among all layers were deleted.

### 3.3. Training auto-encoder model with a CNN architecture

Considering the balance between computational limitation and efficiency, the EANTLI data and composited NPP-VIIRS NTL data were severally split into tiles of 256 × 256 pixels. Since these two NTL data both cover the year of 2013, we built a training set of paired tiles, as $P = \{(X_1, Y_1), (X_2, Y_2), ..., (X_N, Y_N)\}$ by using the 2013 EANTLI data and composited NPP-VIIRS NTL data, respectively, where N indicates the number of tiles. Then this training set was input to train the auto-encoder

model designed above by minimizing the loss function $\mathcal{L}$ between composited NPP-VIIRS NTL data ($Y$) and simulated NPP-VIIRS-like NTL data ($\widehat{Y}$). The loss function adopted in this study is Mean Square Error function and was then optimized by Adam in each deep learning step (Kingma and Ba, 2014). The loss function can be formed as:



$$\mathcal{L} = \frac{1}{N}\sum_{n=1}^{N}\left(\widehat{\boldsymbol{Y}}_n - \boldsymbol{Y}_n\right)^2 \tag{6}$$

### 3.4. Generating an extended time-series of NPP-VIIRS-like NTL data

We generated the extended time-series (2000-2018) of NPP-VIIRS-like NTL data with two components. First, the trained AE model was applied to the 2000-2012 EANTLI data for generating the simulated NPP-VIIRS-like NTL data covering the same period. Second, we composited annual NPP-VIIRS NTL data (2013-2018) from monthly ones. Via appending these two components, the extended time-series (2000-2018) of global NPP-VIIRS-like NTL data were generated.

The split tiles of 2000-2012 EANTLI data with the same size (256 by 256) were input into the trained auto-encoder model to

simulate the NPP-VIIRS-like NTL data. Due to the fluctuation of EVI data, the input EANTLI data have several abnormal pixels which make the output NTL pixels unreasonable. Thus, a post process is required for the simulated NPP-VIIRS-like data, which includes three parts. Firstly, the EANTLI pixels with DN value of 0 were extracted as a dark background mask for each year and overlaid with the simulated NPP-VIIRS-like data to assign the pixels in the same locations as 0, because it is not reasonable to simulate lights at where there is no stable light source. Meanwhile, because of the NPP-VIIRS sensor's

detection limitation, the simulated NTL intensity lower than 1 nano-Wcm$^{-2}$sr$^{-1}$ was also assigned as 0 (Ma et al., 2014). The third post process was designed to make sure the simulated NPP-VIIRS NTL data have the same temporal change as the calibrated DMSP-OLS NTL images. The performance equation was:

$$\mathrm{SNTL}_{(year,i)} = \begin{cases} \mathrm{SNTL}_{(year+1,i)} & \mathrm{NTL}_{(year,i)} > \mathrm{NTL}_{(year+1,i)} \cap \mathrm{SNTL}_{(year,i)} < \mathrm{SNTL}_{(year+1,i)} \\ \mathrm{SNTL}_{(year+1,i)} & \mathrm{NTL}_{(year,i)} < \mathrm{NTL}_{(year+1,i)} \cap \mathrm{SNTL}_{(year,i)} > \mathrm{SNTL}_{(year+1,i)} \\ \mathrm{SNTL}_{(year,i)} & otherwise \end{cases} \tag{7}$$

where $\mathrm{SNTL}_{(year,i)}$ and $\mathrm{NTL}_{(year,i)}$ indicates the simulated NPP-VIIRS-like intensity and calibrated DMSP-OLS NTL

intensity of $i$th pixel in $year$ (from 2000 to 2012).

### 4.  Results

### 4.1. Training of the auto-encoder model with CNN

In the training process of AE model, the learning rate in this study was initialized as 1e-4, and optimized according to Adam algorithm proposed by Kingma and Ba (2014). For weight initialization, the method proposed by He et al. (2015) was adopted

in this study, instead of the traditional random weights from Gaussian distribution. In Fig. 5, the reconstruction loss of each epoch is illustrated. We can see that with the increase of epoch, the loss value decreases and becomes stable near the loss value of 200. This means that once the epoch is larger than some kind of threshold, keeping iteratively training this model is not helpful to improve the accuracy. Therefore, the epoch number is fixed as 4000 based on this trial-and-error experiments.

(Insert Figure 5 near here)



## 4.2. Accuracy evaluation

According to the pixel-level and city-level validation between extended NPP-VIIRS-like NTL data and composited NPP-VIIRS NTL data of 2012 (Fig. 6), our result is close to the composited NPP-VIIRS NTL data at both spatial scales. At pixel level, more than 1 hundred thousand random pixels were selected as validation points and the coefficient of determination ($R^2$) between our results and composited NPP-VIIRS NTL data of 2012 was 0.87 with Root-Mean-Squared-Error (RMSE) of 2.96 nano-Wcm$^{-2}$sr$^{-1}$, at the significant level of 0.05 (Fig. 6(a)). For city-level validation, the total NTL intensity of 40000 selected cities were adopted as variables and the results shown that the extended NPP-VIIRS-like NTL data has a better performance with $R^2$ of 0.94 and RMSE of 3024.62 nano-Wcm$^{-2}$sr$^{-1}$ (Fig. 6(b)). Most of the dots in these two scatter plots are around the nonbias (1:1) line (red line in Fig. 6), which implies that the extended NPP-VIIRS-like NTL data have a positive 1:1 relationship with the composited NPP-VIIRS NTL data.

(Insert Figure 6 near here)

Among the global validation points, we also selected six sub-sets within six countries (United States, Italy, China, Brazil, South Africa, and Australia) and found out that the accuracy in these countries are all acceptable and has no significant spatial variation (Fig. 7). Brazil has the highest accuracy ($R^2 = 0.86$), followed by the United States ($R^2 = 0.84$). The rest countries all have an accuracy higher than 0.7. Australia has the $R^2$ of 0.79, while Italy and China have the $R^2$ of 0.76 and 0.72, respectively. South Africa also has a good $R^2$ of 0.70. Meanwhile, it is worthy of note that in these six sample countries, the RMSE of each country is very small (from 1.67 to 5.72), which means our results are very similar to the calibrated NPP-VIIRS NTL intensity at pixel level.

(Insert Figure 7 near here)

Fig. 8 shown that our extended NPP-VIIRS-like NTL data have a strong agreement with the DMSP-OLS RNTL data in four years (2000, 2004, 2006, and 2010), which implies that the AE model is suitable for simulating NPP-VIIRS-like NTL data during the entire period. Before 2012, the composited NPP-VIIRS NTL data is not available for the validation, but the DMSP-OLS RNTL data is accessible in some separated years. To validate our results before 2012, we have to use the DMSP-OLS RNTL data as the reference data. The DMSP-OLS RNTL data was calibrated by using pre-flight sensor calibration, but has no actual radiance value. Thus, this validation was conducted at city level and the total NPP-VIIRS-like NTL intensity and DMSP-OLS RNTL intensity of each city were calculated and scattered (Fig. 8). In these four years, all the accuracies ($R^2$) are higher than 0.75 and demonstrate our model does work for this entire time series.

(Insert Figure 8 near here)

Finally, the evaluation of the composited NPP-VIIRS NTL data indicates that it is close to the official annual NPP-VIIRS NTL data in 2015, as shown in Fig. 9, according to a comparison between these two data in 2015 based on 5000 random validation points. In other words, using the median value to composite annual NPP-VIIRS NTL data is reasonable and appropriate. In

Fig. 9(a), the similar distribution of green bars and blue bars in these two histogram plots indicated that the pixel number of composited and official annual NPP-VIIRS NTL data, is close to each other, within each bin of 10 nano-Wcm$^{-2}$sr$^{-1}$. The result illustrates that these two datasets have a significant statistic similarity. Meanwhile, the scatter plot between these two NTL intensity at pixel level (Fig. 9(b)) showed that our composited NPP-VIIRS NTL data has a strong linear relationship ($R^2$=0.85

and slope is close to 1) with the official annual NPP-VIIRS NTL data. The two validation results both proved that the composited annual NPP-VIIRS NTL data generation model based on median value is a reasonable solution.

(Insert Figure 9 near here)

### 4.3. The extended time-series (2000-2018) of global NPP-VIIRS-like NTL data

Fig. 10 shows the global spatial distribution of extended NPP-VIIRS-like NTL in 2012, including three enlarged subplots of

New York, Rome, and Shanghai. From each enlarged subplot (Fig. 10(b)-(d)), our results could provide more information for urbanization evaluation, such as the road network and urban spatial structure. The comparison among these three cities shown that New York have much higher NTL intensity than the other two cities and the pixels with high NTL intensity of Rome and Shanghai were located in downtown area.

The dynamics (2000-2018) of extended NTL intensity along with latitude and longitude were also plotted in Fig. 10(e) and (f).

At the global scale, the region between 20°N to 45°N concentrates the most NTL intensity, while the NTL intensity is much lower in the southern hemisphere. From 2000 to 2010, the region between 30°N to 45°N has no significant change of NTL intensity, but has a great enhancement after 2010. For the region between 15°N to 30°N, the NTL intensity has been increasing during the 15 years, which was mostly caused by the China's development. In the longitudinal direction, one of the NTL intensity peaks within western hemisphere region was mostly located in the USA (from 70°W to 100°W). In eastern hemisphere

region, there are three significant peaks in Europe, India, and China (from west to east). The temporal changes of NTL intensity between 2000 and 2005 are generally slighter than that between 2005 to 2010. But from 2010 to 2015, the NTL intensity has a great intensification, even almost all over the world.

(Insert Figure 10 near here)

Fig. 11 reports the trend of NTL intensity from our extended NPP-VIIRS-like NTL data within each continent. Globally, the

NTL intensity has more than doubled during the last 18 years, from 60 million to almost 150 million nano-Wcm$^{-2}$sr$^{-1}$. At the regional scale, Asia and North America have the most intense NTL intensity increase and also appear to be the main contributors to the global urbanization. Europe and South America as the second group both have a stable but slow increase. In Oceania and Africa, the NTL intensity has no obvious increasement, especially before 2010.

(Insert Figure 11 near here)



## 5. Discussion

### 5.1. Evaluation of spatial patterns of extended NPP-VIIRS-like NTL data in 2012

The extended NPP-VIIRS-like NTL data (Fig. 12(IV)), as well as the composited NPP-VIIRS NTL data (Fig. 12(III)), shows a much obvious spatial variations of NTL intensity and less saturation and blooming issue than the calibrated DMSP-OLS data and EANTLI data (Fig. 12(I) and (II)). In Fig. 12, the calibrated DMSP-OLS data have severe saturation issue and blooming issue in all three selected cities: Shanghai, China, Los Angeles, USA, and Cape Town, South Africa. Through fusing the calibrated DMSP-OLS data and EVI data, the quality of the produced EANTLI data have a significant improvement, but it is still suffering a blooming issue. For example, in the periphery of urban area, there are many pixels should be dark but with low NTL intensity, which makes the lit areas to be much larger than the truth. Compared to the EANTLI data, the composited NPP-VIIRS NTL data and our extended NPP-VIIRS-like NTL data have already got a great improvement. Firstly, the main road network can be easily found from the composited or extended NPP-VIIRS NTL data, which indicates that these two datasets can provide more spatial details of the NTL intensity. Secondly, the urban hierarchy structure is much more clearly than before and each lit area was separated, due to the effective elimination of blooming issue.

(Insert Figure 12 near here)

The fluctuation of extended NPP-VIIRS-like NTL data (solid or dashed line) agrees well with that of the composited NPP-VIIRS NTL data (grey part), from the six profiles cross Shanghai, Los Angeles, and Cape Town in Fig. 13. In these three cities, the general trends of extended NPP-VIIRS-like NTL data have a pretty good agreement with the trends of composited NPP-VIIRS NTL data, especially within the urban core area, even though these extended NPP-VIIRS-like NTL profiles showed a little overestimation. Cross Shanghai, parts of the extended NPP-VIIRS-like NTL profiles have an underestimation by comparing with the composited NPP-VIIRS NTL data. This situation mostly appeared outside the urban core area and within the periphery region (e.g., ID: 80 – 90 in Fig. 13(c)).

(Insert Figure 13 near here)

### 5.2. Evaluation of temporal consistency of the time-series (2000-2018) of NPP-VIIRS-like NTL data

The extended time-series of NPP-VIIRS-like NTL data has a consistent temporal trend by comparing with the time-series of census data and analyzing the range of NTL intensity change nearby the junction point (year of 2012) at global scale and country scale.

Firstly, our NPP-VIIRS-like NTL data have a good agreement with the population trend from 2000 to 2018, as shown in Fig. 14. The census data of the entire world and seven countries were collected from World Bank Open Data (https://data.worldbank.org/). A linear regression model was conducted to compare the log-transformed population and total NTL intensity. The $R^2$ at global scale is 0.84 and the $R^2$ of seven selected countries ranges from 0.60 (in United States) to 0.94



(in China). Totally, this result illustrated that the NPP-VIIRS-like NTL data (2000-2018) have a reasonable temporal trend. And in China, Afghanistan, and Thailand, the extended NTL data have a better performance ($R^2$ is larger than 0.8) of population estimation.

(Insert Figure 14 near here)

Secondly, our extended time-series (2000-2018) of NPP-VIIRS-like NTL data have a smooth temporal change, even nearby
the temporal junction point (year of 2012) between the simulated NPP-VIIRS-like NTL data and composited NPP-VIIRS NTL data. In Fig. 15(a), the total NTL intensity and lit pixels (NTL intensity greater than 1 nano-Wcm$^{-2}$sr$^{-1}$) were measured at global scale from 2000 to 2018. The total NTL intensity and lit pixel increased steadily in the first ten years (2000-2010). From 2010 to 2014, the changes of total NTL intensity and lit pixel both increased much more quickly than before but are still stable, no sudden jumps were found during the period (the rectangular box in Fig. 15). In the last four years (2014-2018), the increases
of these two indices became small and slow. Consequently, our NPP-VIIRS-like NTL data source change from simulated NPP-VIIRS-like NTL data (2000-2012) to composited NPP-VIIRS NTL data (2013-2018) does not cause any unreasonable change of NTL intensity during the entire period at the global scale.

Within the six selected countries, as shown in Fig. 15(b)-(g), the number of lit pixels has a stable trend, while the total NTL intensity has some fluctuations before 2010. From 2007 to 2010, an obvious reduction of NTL intensity was appeared in these
six selected countries, even all over the world. After 2010, most countries were recovered and got an increasement of the NTL intensity. This temporal change of NTL intensity is properly agreement with the worldwide Great Recession (2007-2010). This result proves that our extended NPP-VIIRS-like NTL data can truly find out the NTL intensity increasement or decrement within a country. Meanwhile, we also found that each country has different fluctuation range during the Great Recession. For example, the United States has a great slump, but the decrement in China is slight, as shown in Fig. 15(b) and (f), which is
similar to the conclusion (China is able to withstanding the Great Recession) provided by Wen and Wu (2019). It means that our product has the capability of revealing the details of NTL intensity change, not only telling increase or decrease.

(Insert Figure 15 near here)

### 5.3. Limitations

The artificial light from the oceans is not a stable light source (e.g., ship, offshore oil well), and could bring about a
misunderstanding during the AE model training. Hence, a land area mask was applied to remove the ocean parts, even though this procedure could reduce the ability of detecting information from oceans, such as fishery (Waluda et al., 2008) and boats (Elvidge et al., 2015). After a successful population estimation for evaluating the temporal consistency, we believe our NPP-VIIRS-like NTL data have a good performance in urban research. Meanwhile, more applications are still required to promote this dataset.

The second issue is about the composited NPP-VIIRS NTL data generation based on its median value. We found that some

abnormal pixels with no lit were occurred in the 2012 median annual NPP-VIIRS NTL data. This issue is because the NPP-VIIRS NTL monthly composite data was only available from April, 2012, and in the high latitude regions the NTL data have been impacted (Román and Stokes, 2015;Levin, 2017;Román et al., 2018). In summary, this lacking data (from January to March) and the stray light in summer make the median NTL intensity become bias (lower than the usual level), even become

zero. Thus, the cities or pixels with no lit issue were removed in the validation procedure.

The third one is we only focused on the statistical relationship between DMSP-OLS NTL data and NPP-VIIRS NTL data, without considering some physical parameters (e.g., viewing angles), which indeed could influence the cross-sensor calibrations (Li et al., 2019). For the further research, there are still concerns about the time-series of NTL data with better quality and longer time span.

**6.    Data availability**

The extended time-series (2000-2018) of nighttime light data are freely accessible at https://doi.org/10.7910/DVN/YGIVCD (Chen et al., 2020)

**7.    Conclusions**

An extended time-series (2000-2018) of NPP-VIIRS-like NTL data was produced in this study. This product includes two

parts: the simulated NPP-VIIRS-like NTL data (2000-2012) from DMSP-OLS NTL data and the composited annual NPP-VIIRS NTL data (2013-2018). Compared to the composited NPP-VIIRS NTL data in 2012, our extended NPP-VIIRS-like NTL data show a good accuracy, globally, at the pixel ($R^2$: 0.87, RMSE: 2.96) and city levels ($R^2$: 0.95, RMSE: 3024.62). At the region scale, all countries show an acceptable accuracy. The $R^2$ ranges from 0.70 to 0.86 and RMSE is lower than 6 nano-Wcm$^{-2}$sr$^{-1}$. DMSP-OLS RNTL data in 2000, 2004, 2006, and 2010 were compared with the extended NPP-VIIRS-like NTL

data. All $R^2$ are higher than 0.75, which implies that our model is reliable. These evaluations indicate that our extended NPP-VIIRS-like NTL data has a reasonably good and spatially different quality.

Generally, our extended time-series (2000-2018) of NPP-VIIRS-like NTL data shows a similar spatial pattern as the composited NPP-VIIRS NTL data with a good quality regarding the spatial pattern and temporal consistency. The temporal trend agrees with the population change and the global economy event (e.g., the Great Recession). NTL intensity of our product

does not fluctuate around neighbouring years between the simulated NPP-VIIRS-like NTL data (2000-2012) and composited NPP-VIIRS NTL data (2013-2018).

The extended NPP-VIIRS-like NTL data from 2000 to 2018 can be used to better evaluate and analyze the dynamics of demography and urbanization. For example, we can investigate the urban spatial structure, even their road network, and its temporal dynamic for a long time period. Our proposed NTL dataset is available until 2018 so far, but it can be extended in



the future when the monthly NPP-VIIRS NTL data for the whole year (e.g., 2019) become available.

**Author contributions**

**Zuoqi Chen**: Conceptualization, Methodology, Software, Validation, Formal analysis, Visualization, Writing - Original draft.

**Bailang Yu**: Conceptualization, Methodology, Validation, Formal analysis, Writing - Review & Editing, Funding acquisition.

**Chengshu Yang**: Conceptualization, Methodology, Validation, Writing - Review & Editing. **Yuyu Zhou**: Writing – Review &

Editing, Validation. **Xingjian Qian**: Methodology, Validation. **Congxiao Wang**: Validation, Writing - Review & Editing. **Bin**

**Wu**: Methodology, Formal analysis. **Jianping Wu**: Formal analysis, Resources, Supervision.

**Competing interests**

The authors declare that they have no conflict of interest.

**Acknowledgements**

This research was funded by the National Natural Science Foundation of China (No. 41871331 and 41801343), the China

Postdoctoral Science Foundation (2020M671921), the Major Program of National Social Science Foundation of China

(17ZDA068), and the Fundamental Research Funds for the Central Universities of China.

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



**Table 1: The list of used data in this study**

| Dataset | Source | Role |
|---|---|---|
| EANTLI | Calibrated DMSP-OLS NTL data<br>EVI Data | Input Data in AE model (2000-2013) |
| Composited NPP-VIIRS NTL Data | Monthly NPP-VIIRS NTL data | Reference Data for Validation (2012)<br>Label Data in AE model (2013)<br>part of NPP-VIIRS-like NTL data (2013-2018) |
| DMSP-OLS RNTL data | F12-F15_20000103-20001229_rad_v4<br>F14_20040118-20041216_rad_v4<br>F16_20051128-20061224_rad_v4<br>F16_20100111-20101209_rad_v4 | Reference Data for Validation<br>(2000, 2004, 2006, 2010) |



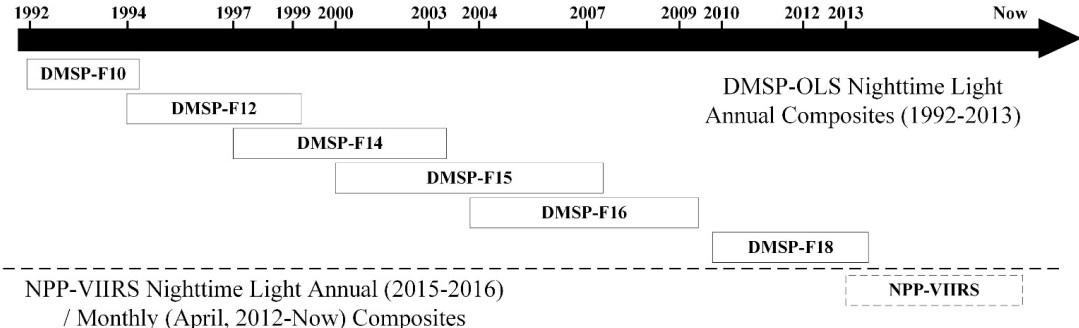


**Figure 1: NTL data from DMSP-OLS and NPP-VIIRS**





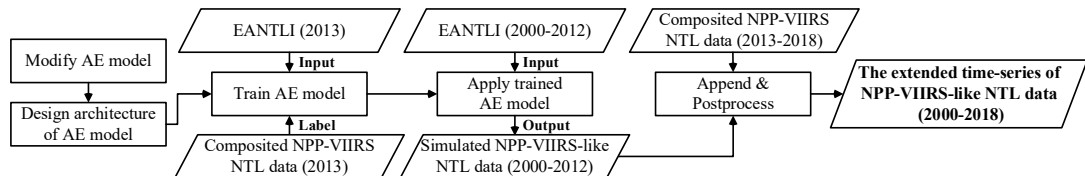

**Figure 2: Flow chart of the generation of NPP-VIIRS-like NTL data.**




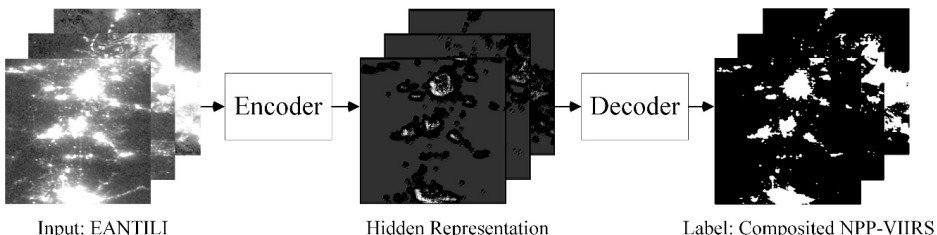

**Figure 3: The framework of the auto-encoder model**



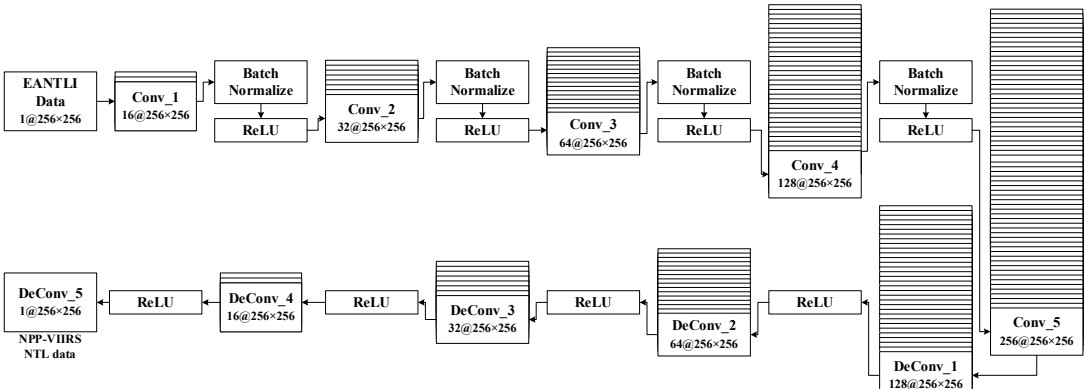

**Figure 4: The overall architecture of our proposed auto-encoder with CNN**

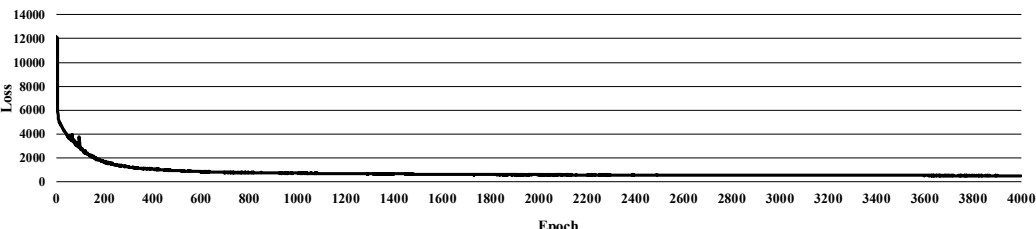

**Figure 5: The training of each epoch's loss value**




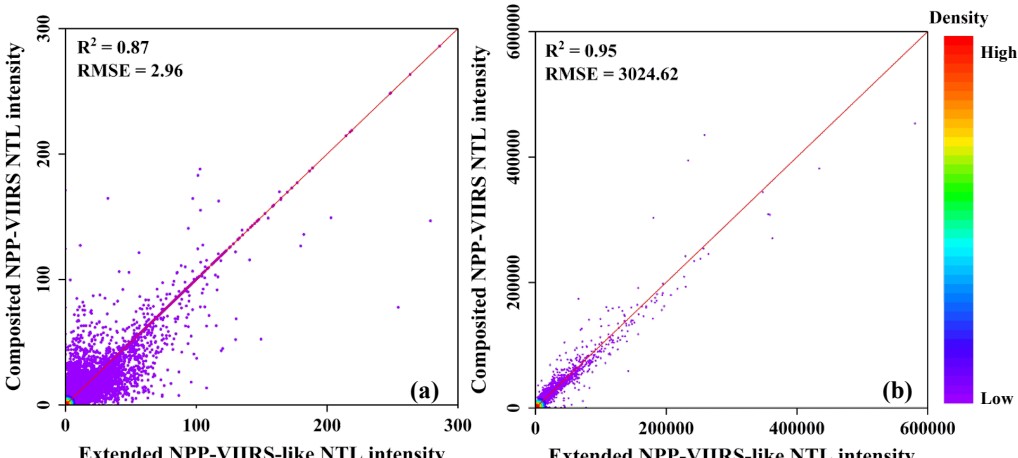

**Figure 6: The comparison between the global composited NPP-VIIRS NTL data and extended NPP-VIIRS-like NTL data in 2012 (unit: nano-Wcm⁻²sr⁻¹): (a) at the pixel and (b) city levels. Solid line denotes the 1:1 line.**

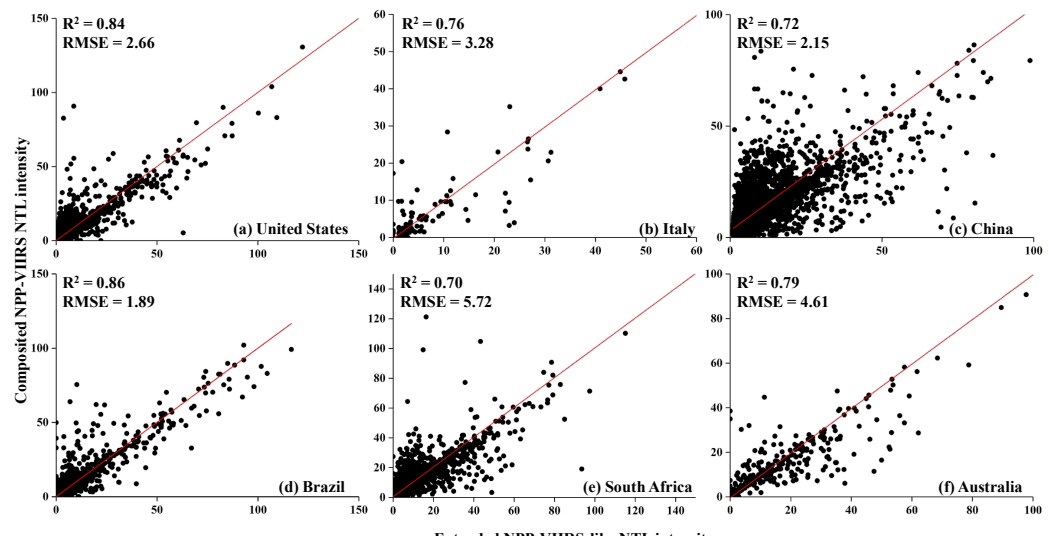

**Figure 7: The comparison between composited NPP-VIIRS NTL data and extended NPP-VIIRS-like NTL data in 2012 (unit: nano-Wcm$^{-2}$sr$^{-1}$) at the pixel level in six countries.**



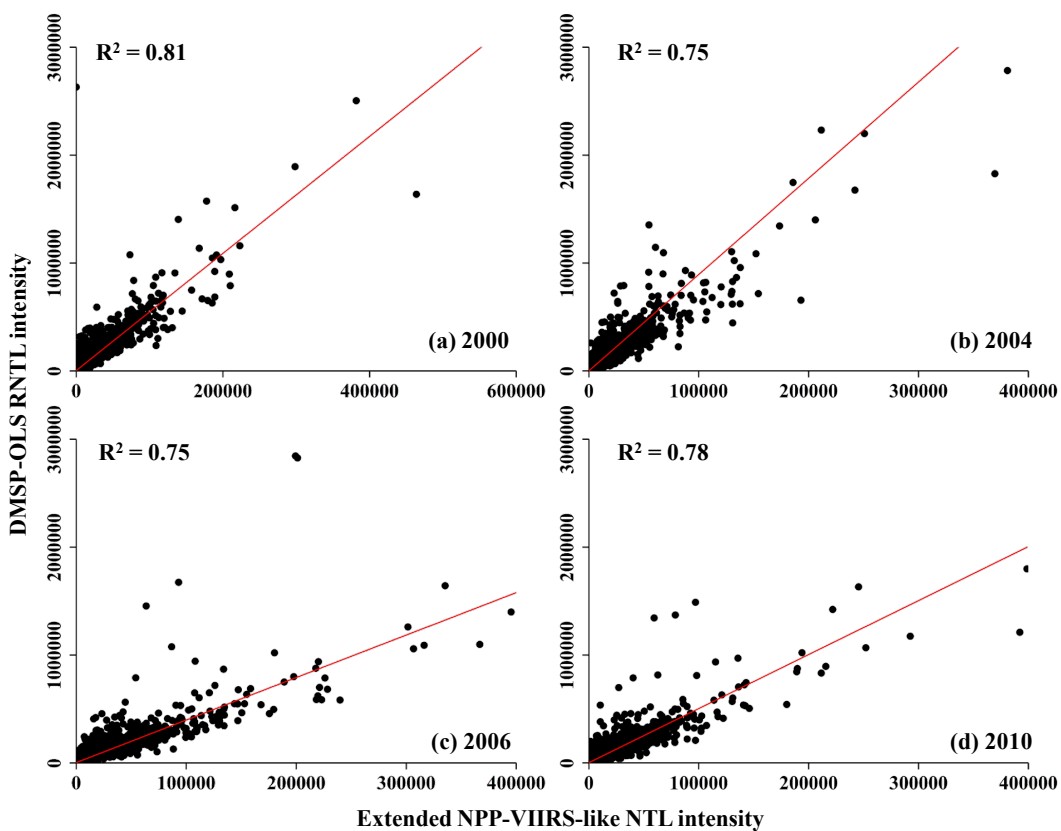

**Figure 8: The comparison between the annual DMSP-OLS RNTL intensity (DN Value) and extended NPP-VIIRS-like NTL intensity**

**(unit: nano-Wcm$^{-2}$sr$^{-1}$) at the city level in (a) 2000, (b) 2004, (c) 2006, and (d) 2010.**



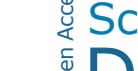

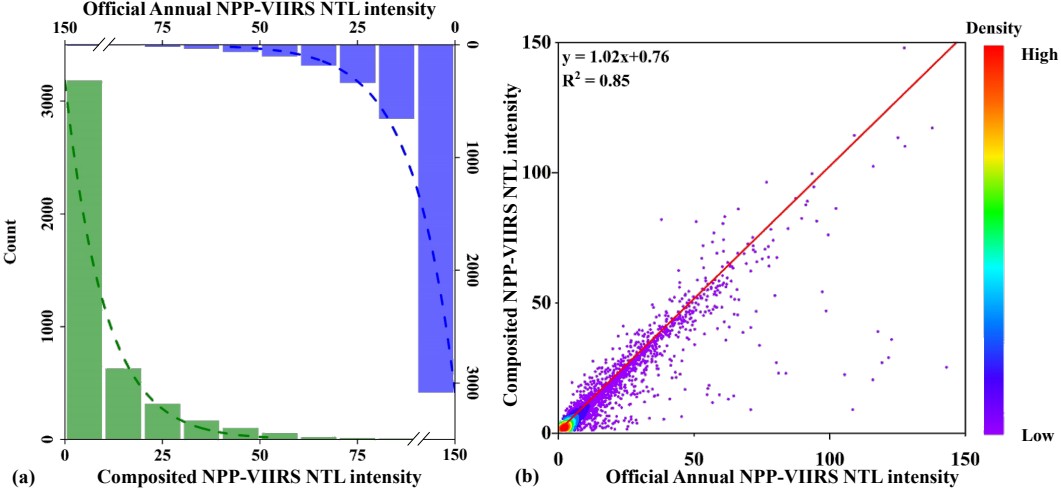

**Figure 9: A comparison of composited and official annual NPP-VIIRS NTL data via (a) histograms and (b) scatter density plot.**



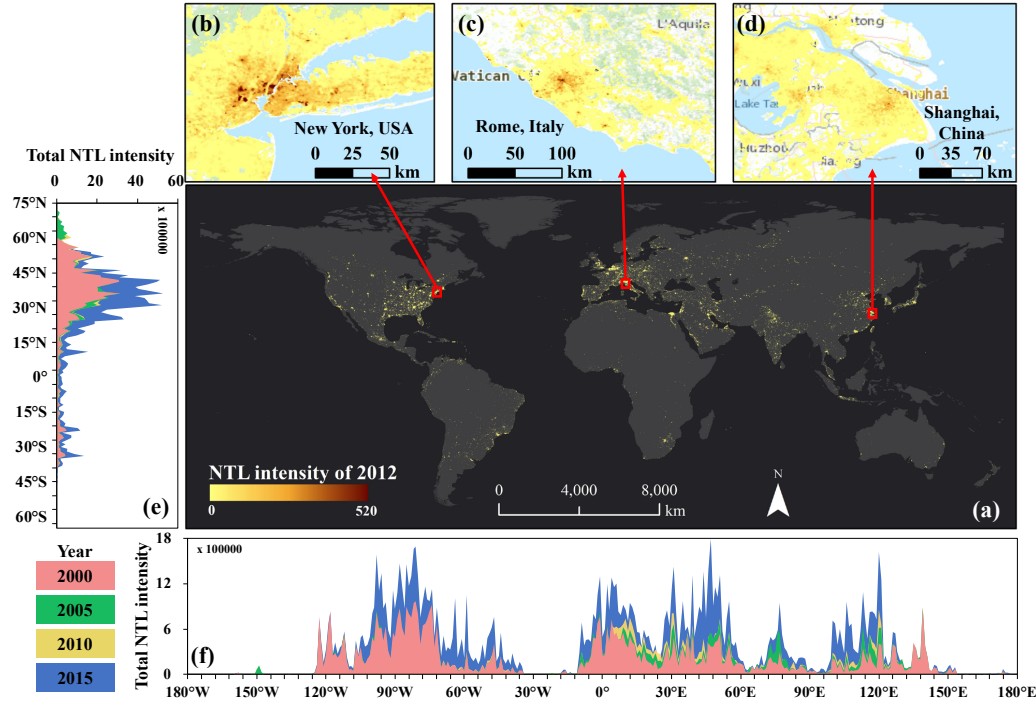

**Figure 10: The extended NPP-VIIRS-like NTL data (unit: nano-Wcm$^{-2}$sr$^{-1}$) with three enlarged subplots of New York, Rome, and Shanghai in 2012 and the dynamics of NTL intensity from 2000 to 2015 by longitude and latitude (1-degree bins).**




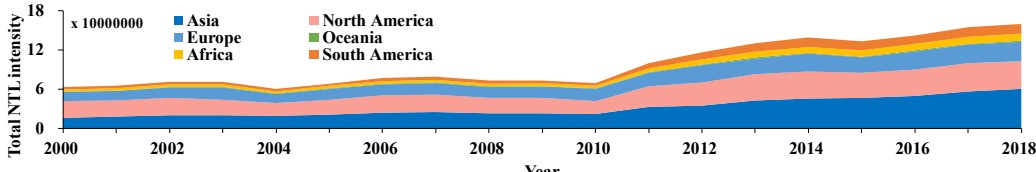

**Figure 11: The dynamics of total NTL intensity from 2000 to 2018 in each continent.**

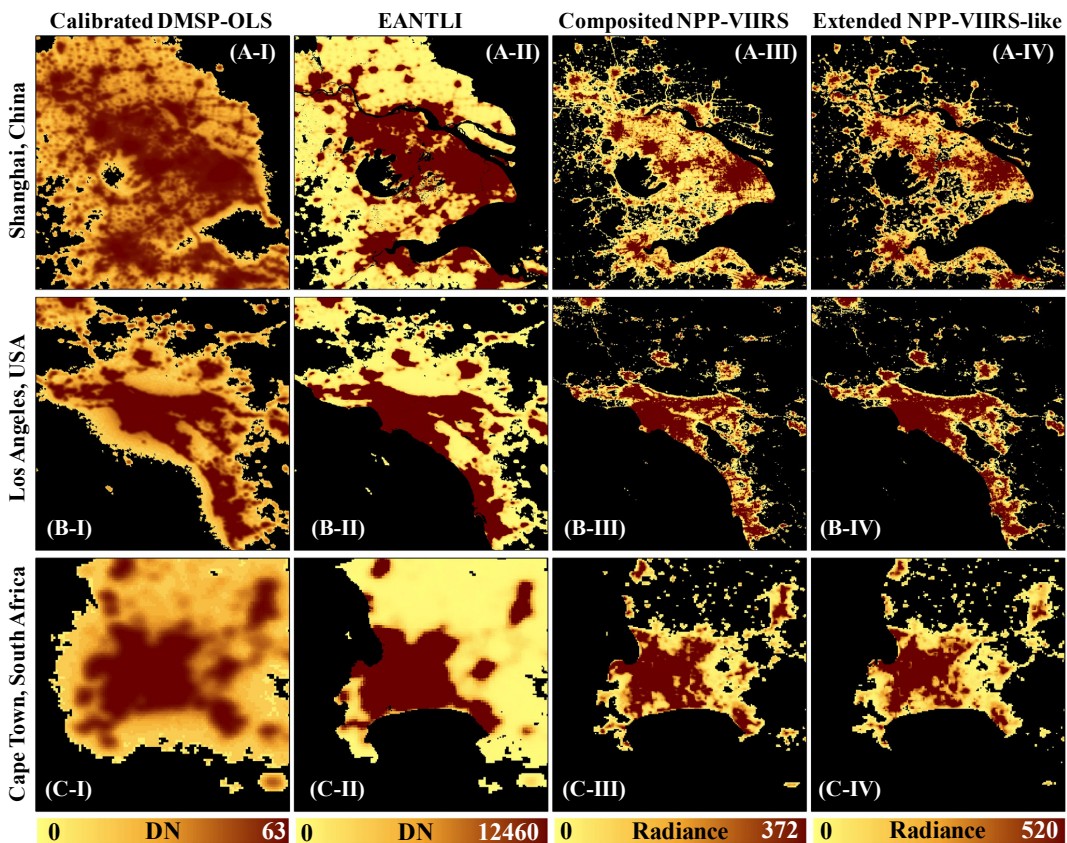

**Figure 12: Spatial patterns of NTL intensity in 2012 of (I) calibrated DMSP-OLS data, (II) EANTL, (III) composited NPP-VIIRS NTL data, and (IV) our extended NPP-VIIRS-like NTL data in three cities (A) Shanghai, China, (B) Los Angeles, USA, and (C) Cape Town, South Africa.**

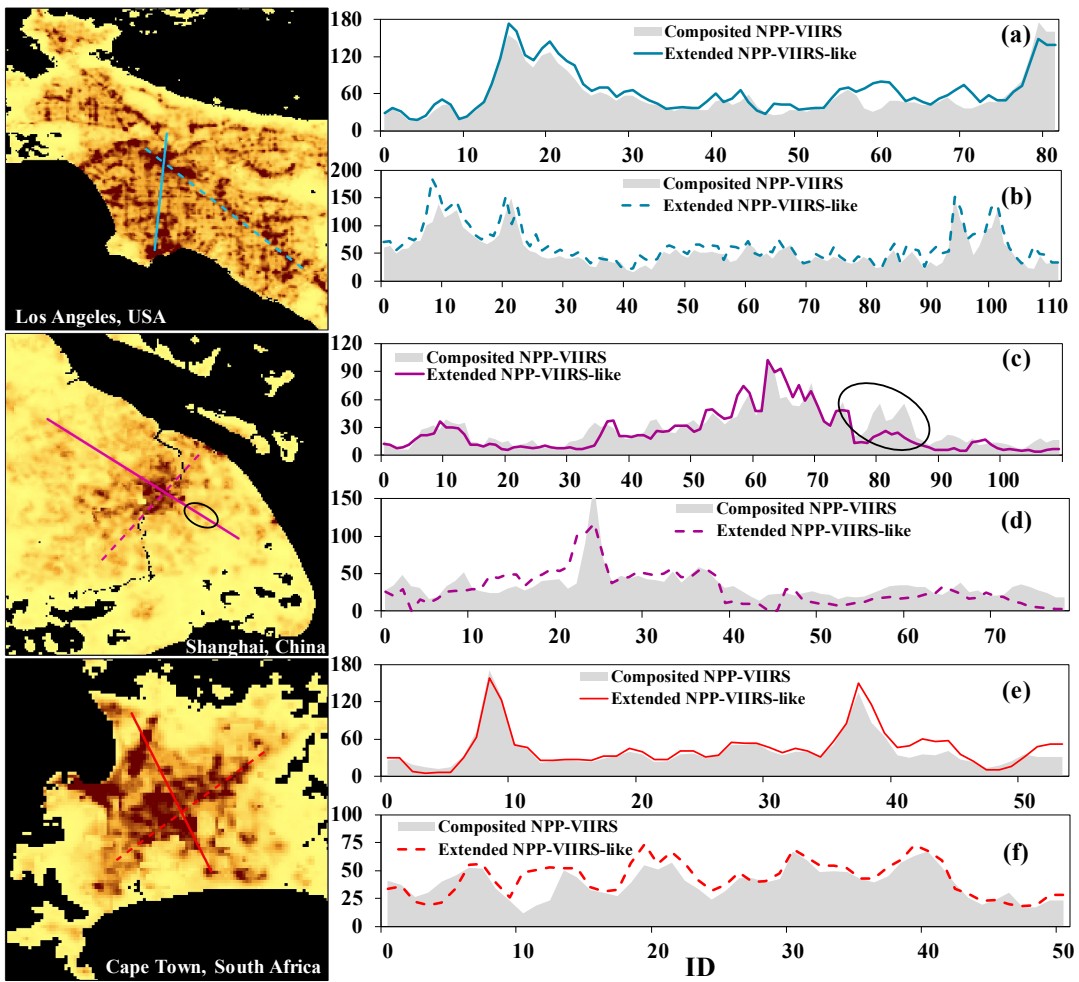

Figure 13: Profiles of composited NPP-VIIRS NTL data and extended NPP-VIIRS-like NTL intensity (unit: nano-Wcm$^{-2}$sr$^{-1}$) across (a and b) Los Angeles, USA, (c and d) Shanghai, China, and (e and f) Cape Town, South Africa.






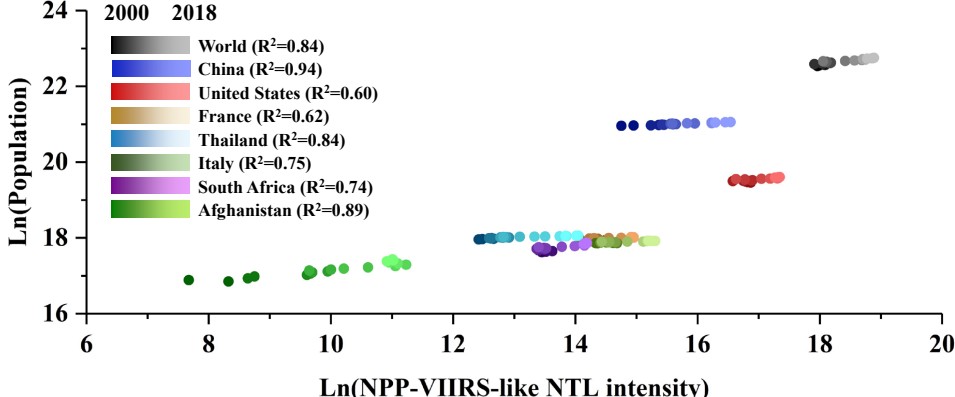

**Figure 14: Comparison of log-transformed urban population with log-transformed total NTL intensity.**



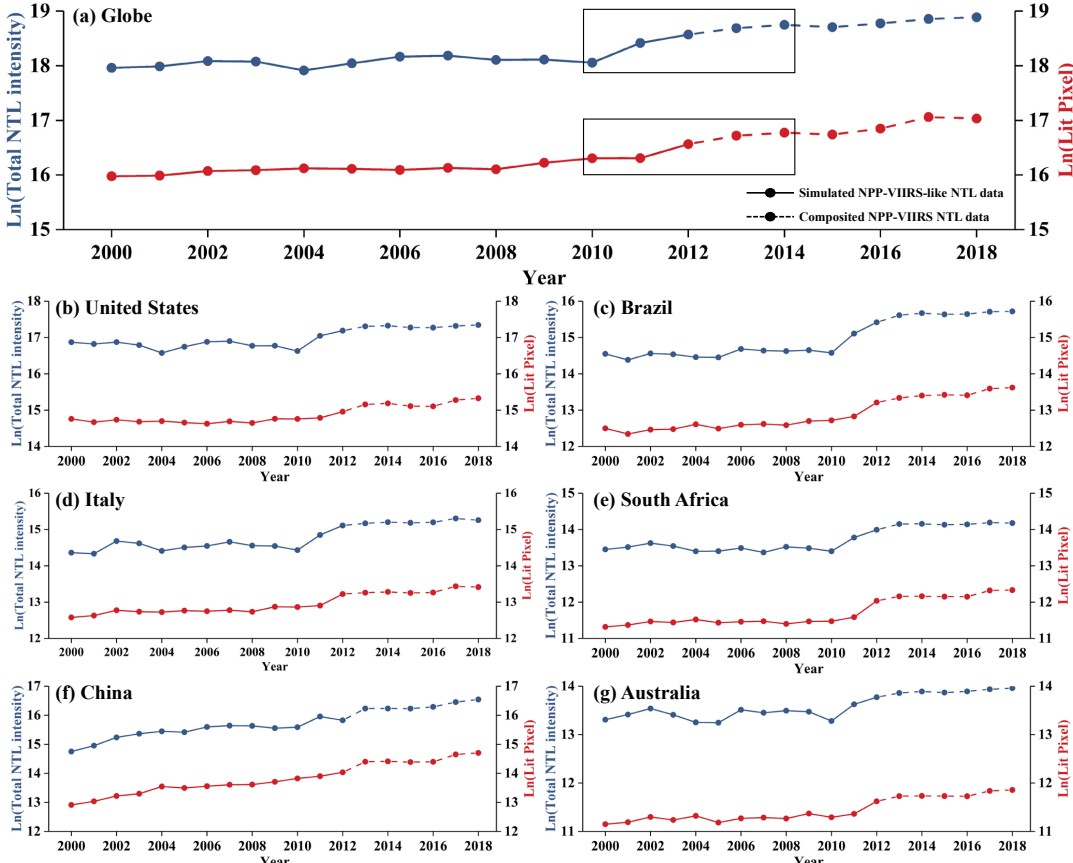

Figure 15: Temporal trend of extended time-series (2000-2018) NPP-VIIRS-like NTL data for the log-transformed total NTL intensity and lit pixels at (a) the global scale and within six selected countries: (b) United States, (c) Brazil, (d) Italy, (e) South Africa, (f) China, and (g) Australia.