# Peer review of "An extended time-series (2000-2018) of global NPP-VIIRS-like nighttime light data from a cross-sensor calibration"

_Earth System Science Data, 2020_

## Referee Comment (RC1) · Anonymous Referee #1 · 30 Nov 2020

Review ESSD-2020-201

Interesting product. Night light (NTL) data used extensively in our science, either from DMSP (annual, only up to 2013) or NPP (authors correctly label this product as NPP-VIIRS): monthly (with some gaps) at higher spatial resolution but available only since 2012. Authors apply an image processing tool - auto-encoder - and several learning / comparison steps to produce a merged annual time series 2000 to 2018. Crossing the sensor divide, e.g. digital number (DN) from DMSP vs radiance values from NPP has proved a difficult barrier up to this point. These authors demonstrate a statistical image comparison tool to cross this gap, significantly different to prior approaches. Some users will prefer higher-res NPP products but for many purposes a verified homogenized longer time span could prove very useful combination. Taking this as an image processing challenge, and developing image-based statistical tools to bridge the DMSP / NPP gap, represents an important and useful approach! Authors have a prepared a good description but one that could benefit substantially from many clarifications and improvements.

Data accessible from a DataVerse site, as 19 separate annual images. My Mac stumbles a bit on the format designator (.tif) vs .tiff but I can open NTL images with GIS and image processing software. Compressed file = 50 MB, single annual image = 9 GB! Authors might add description of file structure and warning about image sizes in the Data Availability section. Authors might consider also making data for one of their figures (e.g. Figure 9 for this reviewer but editor might have different suggestion) available to save users from needing to do our own mass download and compilation?

Overall, well written with reasonable assessments of uncertainty (expressed as correlation coefficients) of various intermediate products and of final composite. Validation / verification mostly consists of comparison at various steps of modified images vs originals. Really not much else one can do with NTL; no comparable alternate sources. (Authors define and use better terms - extended, composited, official - but sometimes as verbs/actions and other times as descriptors/adverbs? Quick search and check should resolve any language uncertainties.) However, this statement (lines 363 and 364 in Limitations section): "For the further research, there are still concerns about the time-series of NTL data with better quality and longer time span." Paper cited in the sentence before (Li et al. 2019) focused only on NPP data, so how does their work impact this product? What - in terms of uncertainty - does that sentence portend? Better calibrations? Longer time-series? Corrections based on viewing angles would give better correlation coefficients? A future version that renders the present version obsolete? Please provide explanation and assurance. I suspect authors mean something like: further work e.g. on viewing angles might result in (slight?, substantial?) improvements in correlations. As written however the sentence leaves a reader / user wondering about product as described? A so-called 'harmonized DMSP' NTL data product already exists (Scientific Data, https://www.nature.com/articles/s41597-020-0510-y), discussed by these authors in Introduction at lines 66 to 73) but this reader, while impressed with this approach and this product, never found a quantitative statement of improvement over prior work?

Section 2, Data

Please provide exact data sources. Even if data came from Google Earth Engine (referenced earlier) authors must provide exact DOI, version number, date of access, etc. A data source table would work very well here. Something like Table 1 but with much more detail. Authors do a good job of describing (and labeling) their source products and their own subsequent modifications but user needs to see exact starting points. Why Colorado School of Mines? Whatever the source, assure open accessibility and provide exact details.

Did authors use Google Earth Engine to access data or as a computing service?

Line 145: Some prior reason for selecting those three cities?

Line 165: Figure 2 does not represent full processing scheme which includes access, pre-processing, AE CNN training, and post-processing. Figure 2 includes post-processing but omits substantial necessary pre-processing? Starts instead at AE step.

Line 190, Figure 4: informative perhaps as description of an image processing tool but less helpful in a data description of new homogenized NTL product? Authors / editors need to distinguish between focus on AE CNN tool and description of useful NTL data product.

Line 207: "Adam"?? Reader does not learn until line 228 that 'Adam' refers to an already published processing algorithm.

Line 231: 'epoch' as used here refers not to geologic time period but to an iteration quanta of the AE deep learning tool? At some point the loss term approaches some statistical noise lower limit? Authors mention (line 233) 4000 as their empirically-chosen iteration limit but this also has implications for computing resources? Compressed vertical scale in Fig 5 not helpful here.

Line 234, Figure 5: Same question as for line 190 above: does this relate to new image processing tool or to higher quality NTL product?

Line 235 and following, section 4.2 'Accuracy evaluation'. Confusion here about numbers and Figure 6. Fig 6a has ordinate and abscissa range of 300, suggesting that an individual pixel can have an intensity value up to e.g. 250 or 300? From text, Fig 6a should include 100k points? Perhaps give us an n(umber) value? Fig 6b should represent 400000 cities with max NTL intensity values of 600k (e.g. a range of 0 to 600k)? Integrated above some minimum intensity value over a NTL-determined or geopolitically-defined city area? The density scale (population density?) on the right refers only to 6b?

Line 253 Figure 7: same units as for Fig 6 but now for country. Authors discuss correlations (agree, good in all cases) but changing axis extents/scales indicate USA comparable to Brazil, both greater than South Africa which is greater (brighter) than China and Australia both of which brighter than Italy. Confirm from other evidence? Or, if we should not indulge in quick visual speculations, use a uniform axis scale on all panels. Again, knowing n(umber) could prove very useful.

Line 262 Figure 8: What city or cities. Same 400k cities used in Fig 6b? DN vs nanoWatt intensities so the axis scales have changed, correlation coefficients as before but now the solid red lines indicate data slopes not 1:1 lines. What do we make of the changing slopes. No mention from authors. We need n(umber) indication! 2004 tends to show outlying data points below the slope line, e.g. high intensities for low DN. 2006 and 2010 show opposite pattern: higher DN for lower intensities. Useful? Not useful? Perhaps slope lines have no value outside of specific chart but the apparent temporal variance of those lines 2000 to 2010 does not accord with Figure 10?

Line 272 Figure 9: Authors justify use of median, but from where do these data come? Text says "based on 5000 random validation points." Random pixels? Apparently so, with axis scales at 0 to 150. All with data greater than 0, e.g. density of 'low' or greater? Density axis (again, population density?) refers in this case to city, not pixel? What do I not understand here?

Line 278: the term "downtown" conveys neither geographic nor NTL density accuracy or specificity.

Figure 10 very useful, lots of information. Text discussion of Fig 10 implies greatest NTL intensity change occurs globally over 2010 to 2015? Highest longitudinal peak might occur in Middle East (not = Europe), not mentioned in text?

Line 316, Figure 13: Potentially useful figure but quite confusing. ID units relative, not km? No explanation for the circled discrepancy in 13c? Values represent averages (means, medians?) along track at some spatial resolution. Or pixel by pixel? Y-axis scales change with every city; use a common scale to allow city-by-city comparisons?

Line 319: "nearby the junction point" what does this mean? End of simulated and start of composited?

Line 323: world bank population data. We need a version number, DOI, accurate references, assurance of open access, … This data product should be referenced in improved Table 1, above?

Line 328 Figure 14. Ln:Ln plot gives little to no confidence. Linear trend of log-transformed data, is that even valid? Use an inset to show global data in separate non-log figure? Once reader gains confidence with global numbers, regional numbers will make more sense. Or separate into two panels, one for global and one for 7 countries. If this represents first use of population data, then what did density scales in Figs 6 and 9 represent? Some spatial clustering of high intensity pixels? What?

Line 330: here a reader learns about "temporal junction point" and can use Figure 15 to understand. Need similar clarity at line 319.

Lines 334 to 336: Do the time-dependent changes discussed here, strong from 2010 to 2014 but diminished from 2014 to 2018, accord with what the authors concluded earlier, e.g. about changes globally and regionally from 2010 to 2015 as in Fig 10?

Line 346 Figure 15. Same six countries used in Fig 7? Why these six? Y-axis scale varies in both minimum and maximum in almost every plot. Ln values? Not helpful! Again, one scale for global and a second uniform scale for all countries? These apparent offsets per country with time explain the different slopes in Figure 7?

Small language issues throughout (e.g. line 49 comparable not comparability, "worthy to note" rather than noteworthy in several places, etc.); one hopes / assumes Copernicus pre-publication efforts will clean these up?

---

## Referee Comment (RC2) · Anonymous Referee #2 · 23 Dec 2020

This paper introduces an extended time-series (2000-2018) of NPP-VIIRS-like NTL data through a cross-sensor calibration from DMSP-OLS NTL data (2000-2012) and a composition of monthly NPP-VIIRS NTL data (2013-2018). The authors claim that compared with the annual composited NPP-VIIRS 20 NTL data in 2012, their product of extended NPP-VIIRS-like NTL data shows a good consistency at the pixel and city levels with R2 of 0.87 and 0.95, respectively. The paper is well organized and written clearly, and the data set should be of interest to users who use the NTL data. Therefore, the paper should be accepted for publication after the follow issue is addressed.

It is strongly recommended that a short description of the uniqueness of the method

used should be included in the abstract. The current abstract only tells the reader that there is a new product and it is better, but it fails to say what it is and how it is produced. For example, if the uniqueness is that they relied on the vegetation index adjusted NTL to perform the simulation, it should say so up front in the abstract, so that readers can get the main point without having to read the entire paper. Other than that, the paper is well written.

Thanks.

---

## Author Comment (AC1) · 19 Jan 2021

The comment was uploaded in the form of a supplement:
https://essd.copernicus.org/preprints/essd-2020-201/essd-2020-201-AC1-supplement.pdf

---

## Author Response (AR1)

**Response to Review ESSD-2020-201-RC1**

Interesting product. Night light (NTL) data used extensively in our science, either from DMSP (annual, only up to 2013) or NPP (authors correctly label this product as NPP-VIIRS): monthly (with some gaps) at higher spatial resolution but available only since 2012. Authors apply an image processing tool - auto-encoder - and several learning / comparison steps to produce a merged annual time series 2000 to 2018. Crossing the sensor divide, e.g. digital number (DN) from DMSP vs radiance values from NPP has proved a difficult barrier up to this point. These authors demonstrate a statistical image comparison tool to cross this gap, significantly different to prior approaches. Some users will prefer higher-res NPP products but for many purposes a verified homogenized longer time span could prove very useful combination. Taking this as an image processing challenge, and developing image-based statistical tools to bridge the DMSP / NPP gap, represents an important and useful approach! Authors have a prepared a good description but one that could benefit substantially from many clarifications and improvements.

**Response:** Thank you very much for your constructive comments. We improved our paper based on your comments and suggestions. Please see the detailed point-to-point response below.

**Comment 1:** Data accessible from a DataVerse site, as 19 separate annual images. My Mac stumbles a bit on the format designator (.tif) vs .tiff but I can open NTL images with GIS and image processing software. Compressed file = 50 MB, single annual image = 9 GB! Authors might add description of file structure and warning about image sizes in the Data Availability section. Authors might consider also making data for one of their figures (e.g. Figure 9 for this reviewer but editor might have different suggestion) available to save users from needing to do our own mass download and compilation?

**Response:** Each year's NTL image was generated as a standard GeoTIFF image and compressed as a Zip file. We totally published 19 zip files (from 2000 to 2018). In each zip file, the data contains five files, including LongNTL_year.tif, LongNTL_year.tif.aux.xml, LongNTL_year.tif.ovr, LongNTL_year.tif.xml, and LongNTL_year.tfw. We also added the details of data spatial reference and spatial resolution in the manuscript.

Meanwhile, the data of Figure 8 (original Figure 9) and Figure 14 (original Figure 15) have both been submitted to HARVARD Dataverse and are freely accessible now from the same link (https://doi.org/10.7910/DVN/YGIVCD).

To make users better understand our dataset, we have modified the statement in Data availability section, as "The extended time-series (2000-2018) of nighttime light data in WGS84 coordinate system with a spatial resolution of 15 arc-second (~500 meter) can be freely accessed at https://doi.org/10.7910/DVN/YGIVCD (Chen et al., 2020), which is stored as Zip file (~50MB) for each year. By uncompressing the zip file, the annual NPP-VIIRS-like NTL data is provided in GeoTIFF format (~ 9GB). These data can be processed using open-source software such as QGIS. We also included two data tables as Microsoft Excel XLSX files. One contains 40,000 sample points for comparing composited and official annual NPP-VIIRS NTL data in Figure 8, and the other is the data used for the temporal trend

analysis of our extended time-series of NPP-VIIRS-like NTL data in Figure 14."

**Comment 2:** Overall, well written with reasonable assessments of uncertainty (expressed as correlation coefficients) of various intermediate products and of final composite. Validation / verification mostly consists of comparison at various steps of modified images vs originals. Really not much else one can do with NTL; no comparable alternate sources. (Authors define and use better terms - extended, composited, official - but sometimes as verbs/actions and other times as descriptors/adverbs? Quick search and check should resolve any language uncertainties.) However, this statement (lines 363 and 364 in Limitations section): "For the further research, there are still concerns about the time-series of NTL data with better quality and longer time span." Paper cited in the sentence before (Li et al. 2019) focused only on NPP data, so how does their work impact this product? What - in terms of uncertainty - does that sentence portend? Better calibrations? Longer time-series? Corrections based on viewing angles would give better correlation coefficients? A future version that renders the present version obsolete? Please provide explanation and assurance. I suspect authors mean something like: further work e.g. on viewing angles might result in (slight?, substantial?) improvements in correlations. As written however the sentence leaves a reader / user wondering about product as described? A so-called 'harmonized DMSP' NTL data product already exists (Scientific Data, https://www.nature.com/articles/s41597-020-0510-y), discussed by these authors in Introduction at lines 66 to 73) but this reader, while impressed with this approach and this product, never found a quantitative statement of improvement over prior work?

**Response:** Thanks for your comments and suggestions. First, we have checked the terms of extended, composited, official, and made revisions where needed to avoid the confusion.

Second, Li et al. (2019) and (Román et al., 2018) have mentioned that the viewing angle and lunar zenith angle of NPP-VIIRS NTL data could influence its data quality, so we inferred that such physical parameters could also affect our cross-sensor calibration and the data quality. To clarify the statement, we revised the sentences in section 5.3 (around line 371-376) as "Meanwhile, introducing physical parameters during the pre-processing of data could probably further improve our extended NTL product because it has been proven that physical parameters such as viewing angle (Li et al., 2019) and lunar zenith angle (Román et al., 2018) could influence the NTL data quality. However, due to the limitation of data accessibility, the land cover/land use data and physical parameters were not involved in this study."

Li, X., Ma, R., Zhang, Q., Li, D., Liu, S., He, T., and Zhao, L.: Anisotropic characteristic of artificial light at night – Systematic investigation with VIIRS DNB multi-temporal observations, Remote Sens Environ, 233, 111357, doi: 10.1016/j.rse.2019.111357, 2019

Román, M. O., Wang, Z., Sun, Q., Kalb, V., Miller, S. D., Molthan, A., Schultz, L., Bell, J., Stokes, E. C., Pandey, B., Seto, K. C., Hall, D., Oda, T., Wolfe, R. E., Lin, G., Golpayegani, N., Devadiga, S., Davidson, C., Sarkar, S., Praderas, C., Schmaltz, J., Boller, R., Stevens, J., Ramos González, O. M., Padilla, E., Alonso, J., Detrés, Y., Armstrong, R., Miranda, I., Conte, Y., Marrero, N., MacManus, K., Esch, T., and Masuoka, E. J.: NASAs Black Marble nighttime lights product suite, Remote Sens Environ, 210, 113 -143, doi: 10.1016/j.rse.2018.03.017, 2018

Third, the 'global harmonized DMSP NTL data' is a DMSP-OLS-like NTL data and has a better temporal

consistency, by harmonizing the NPP-VIIRS NTL to DMSP-OLS-like data. This product recorded the simulated DMSP-OLS-like digital number (DN) value from 1992 to 2018, while our extended NPP-VIIRS-like NTL data recorded the radiance value from 2000 to 2018. Since these two datasets have totally different unit, even the physical implications, the direct comparison between them is not reasonable. This is also the reason why we used the composited NPP-VIIRS NTL data, rather than the 'global harmonized DMSP NTL data', as the reference data to evaluate our extended NPP-VIIRS-like NTL data in our manuscript.

**Comment 3:** Section 2, Data. Please provide exact data sources. Even if data came from Google Earth Engine (referenced earlier) authors must provide exact DOI, version number, date of access, etc. A data source table would work very well here. Something like Table 1 but with much more detail. Authors do a good job of describing (and labeling) their source products and their own subsequent modifications but user needs to see exact starting points. Why Colorado School of Mines? Whatever the source, assure open accessibility and provide exact details.

**Response:** Thanks for your suggestion. We improved Table 1 to as suggested.

**Table 1: The list of used data in this study**

| Dataset | Source | Role |
|---|---|---|
| EANTLI | Calibrated DMSP-OLS NTL data[1]
 EVI Data[2] | Input data in AE model (2000-2013) |
| Composited NPP-VIIRS NTL Data | Monthly NPP-VIIRS NTL data[3] | Reference data for validation (2012)
 Label data in AE model (2013)
 part of NPP-VIIRS-like NTL data (2013-2018) |
| DMSP-OLS RNTL Data[3] | F12-F15_20000103-20001229_rad_v4
 F14_20040118-20041216_rad_v4
 F16_20051128-20061224_rad_v4
 F16_20100111-20101209_rad_v4 | Reference data for validation (2000, 2004, 2006, 2010) |
| Census Data[4] | Total Population (ID: SP.POP.TOTL) | Reference data for temporal consistency validation (2000-2018) |

[1.] Accessed from Li et al. (2020) in February 2020.

[2.] Accessed from MOD13A1 version 5 based on Google Earth Engine in May 2020.

[3.] Accessed from the Earth Observations Group (EOG) in Colorado School of Mines (https://payneinstitute. mines.edu/eog/nighttime-lights/) in May 2020.

[4.] Accessed from World Bank (2020) in May 2020.

The statement in section 2.3 (around line 161-162) has been modified as "The four selected DMSP-OLS RNTL data were accessed from Earth Observations Group (EOG) in Colorado School of Mines, as shown in Table 1."

Li, X., Zhou, Y., Zhao, M., and Zhao, X.: A harmonized global nighttime light dataset 1992–2018, Scientific Data, 7, 168, doi: 10.1038/s41597-020-0510-y, 2020.

World Bank: Population, total, World Development Indicators, https://data.worldbank.org/indicator/SP.POP.TOTL, 2020.

**Comment 4:** Did authors use Google Earth Engine to access data or as a computing service?

**Response:** We used Google Earth Engine to generate the global EVI data. In the section 2.1 (around line 121), the statement was modified as "the MOD13A1 EVI products were processed as an annual average EVI (Jing et al., 2015) using Google Earth Engine."

**Comment 5:** Line 145: Some prior reason for selecting those three cities?

**Response:** According to Shi et al., (2014b), the NTL intensity in other areas should not exceed the maximum NTL intensity in the center of large cities. If the pixel has a value larger than the maximum, the pixel was identified as an abnormal pixel and needs be adjusted. Thus, we needed to select the large cities to represent these maximum values.

The cities were selected based on the city rank by GaWC 2020 (Globalization and World Cities in 2020). First, New York and London with the alpha++ (the highest rank) level were selected. Then, Shanghai and Beijing with the alpha+ level in China were selected. In this way, these four cities can generally capture the maximum values to process the abnormal pixels.

In section 2.2, the statement around line 146-151 has been modified as "Then, we assumed that the NTL intensities in other areas do not exceed the maximum NTL intensity in the center of large cities. If the pixel has a value larger than the maximum, this pixel was identified as the abnormal pixel and was adjusted (Shi et al., 2014b). According to the global city rank from the Globalization and World Cities (GaWC) Research Network (Taylor et al., 2010), two cities (New York City in USA and London in United Kingdom) with the alpha++ level and the two largest cities in China (Shanghai and Beijing) were selected in this study to capture the maximum NTL intensity as the threshold to adjust the abnormal pixels for each year."

Shi, K., Yu, B., Huang, Y., Hu, Y., Yin, B., Chen, Z., Chen, L., and Wu, J.: Evaluating the Ability of NPP-VIIRS Nighttime Light Data to Estimate the Gross Domestic Product and the Electric Power Consumption of China at Multiple Scales: A Comparison with DMSP-OLS Data, Remote Sensing, 6, 1705-1724, 10.3390/rs6021705, 2014b.

**Comment 6:** Line 165: Figure 2 does not represent full processing scheme which includes access, pre-processing, AE CNN training, and post-processing. Figure 2 includes post-processing but omits substantial necessary pre-processing? Starts instead at AE step.

**Response:** Thank you for your suggestion. We have re-drawn our flow chart of the generation of extended NPP-VIIRS NTL data and made a corresponding slight change in Figure 2.

[Figure]

**Figure 2: Flow chart of the generation of NPP-VIIRS-like NTL data.**"

**Comment 7:** Line 190, Figure 4: informative perhaps as description of an image processing tool but less helpful in a data description of new homogenized NTL product? Authors / editors need to distinguish between focus on AE CNN tool and description of useful NTL data product.

**Response:** This CNN architecture is what we used to capture the relationship between EANTLI and NPP-VIIRS NTL data. In case the readers want to repeat our experiment or learn from our experiment to reconstruct other images, this figure can help them build their own CNN architecture quickly. So we still kept this figure in the manuscript.

**Comment 8:** Line 207: "Adam"?? Reader does not learn until line 228 that 'Adam' refers to an already published processing algorithm.

**Response:** thank you for pointing this out. We modified the statement in section 3.4, line 211-212 (original line 207) as "The loss function adopted in this study is the Mean Square Error function and was then optimized by the Adam algorithm proposed by Kingma and Ba (2014) in each deep learning step." We also modified the statement in section 4.1, line 233-234 (original line 228) as "In the training process of the AE model, the learning rate in this study was initialized as 1e-4 and optimized by using the Adam algorithm."

**Comment 9:** Line 231: 'epoch' as used here refers not to geologic time period but to an iteration quanta of the AE deep learning tool? At some point the loss term approaches some statistical noise lower limit? Authors mention (line 233) 4000 as their empirically-chosen iteration limit but this also has implications for computing resources? Compressed vertical scale in Fig 5 not helpful here.

**Response:** Thanks for your comments. "epoch" is a deep learning related terminology and refers to how many iterations were executed. Computing resources should not affect the number of iterations, but affect

the cost time. As suggested in Comment 10, Figure 5 was removed from the manuscript. The corresponding statement in section 4.1 (around line 234-239) was revised as "For weight initialization, this study employed the method proposed by He et al. (2015) instead of the traditional random weights from Gaussian distribution. Theoretically, the AE model was iteratively trained until the reconstruction loss became stable. In this study, the loss value tended to be stable around 200 when the number of training iterations reached 4000, which implies that the increase of iterations beyond 4000 cannot further improve the model precision."

**Comment 10:** Line 234, Figure 5: Same question as for line 190 above: does this relate to new image processing tool or to higher quality NTL product?

**Response:** Figure 5 presented the changing loss value during the iterative training. As it is not related to the image process or data enhancement, it was removed in this version. The corresponding statement in section 4.1 (around line 234-239) was revised as "For weight initialization, this study employed the method proposed by He et al. (2015) instead of the traditional random weights from Gaussian distribution. Theoretically, the AE model was iteratively trained until the reconstruction loss became stable. In this study, the loss value tended to be stable around 200 when the number of training iterations reached 4000, which implies that the increase of iterations beyond 4000 cannot further improve the model precision."

**Comment 11:** Line 235 and following, section 4.2 'Accuracy evaluation'. Confusion here about numbers and Figure 6. Fig 6a has ordinate and abscissa range of 300, suggesting that an individual pixel can have an intensity value up to e.g. 250 or 300? From text, Fig 6a should include 100k points? Perhaps give us an n(umber) value? Fig 6b should represent 400000 cities with max NTL intensity values of 600k (e.g. a range of 0 to 600k)? Integrated above some minimum intensity value over a NTL-determined or geopolitically-defined city area? The density scale (population density?) on the right refers only to 6b?

**Response:** Thank you for your questions. For the pixel-level accuracy evaluation, 150,000 random pixels were selected with a range of NTL intensity from 0 to 300. For the city-level accuracy evaluation, we calculated each city's NTL total intensity for 40,000 cities across the world. The NTL total intensity with a range of 0 to 600,000, refers to the sum of all pixels' NTL intensity within an administrative boundary. For the terms of density scale, it is the kernel density of the dots in scatter plot, rather than population density. Following the color rule of the density scale, the dots were colored according to their kernel density. The higher the density is, the warmer color the dot shows on the plots. It is seen from the figure that the dots with higher density are clustered around the origin of coordinates, where the dots are mostly colored red.

To clarify, Figure 5 (original Figure 6) has been redesigned and the statement in section 4.1 (around line 241-247) was revised as "At the pixel level, 150,000 random pixels were selected as validation points and the coefficient of determination ($R^2$) between our results and composited NPP-VIIRS NTL data of 2012 was 0.87 with Root-Mean-Squared-Error (RMSE) of 2.96 nano-Wcm$^{-2}$sr$^{-1}$, at the significant level. The dots were colored based on their kernel density, which follows the color ramp of the density scale. A warmer colored dot represents a higher density. It can be observed that the dots were mostly clustered around the low NTL intensity (the origin of coordinates). For the city-level validation, the total NTL intensity for each city (i.e., the sum of all pixels' NTL intensities within each of 40,000 cities) was adopted as the variable and the results showed that the extended NPP-VIIRS-like NTL data has a better performance with $R^2$ of 0.94 and RMSE of 3024.62 nano-Wcm$^{-2}$sr$^{-1}$ (Fig. 5(b)).

[Figure]

**Figure 5: The comparison with kernel density between the global composited NPP-VIIRS NTL data and extended NPP-VIIRS-like NTL data in 2012 (unit: nano-Wcm$^{-2}$sr$^{-1}$): (a) at the pixel and (b) city levels. Solid line denotes the 1:1 line and N is the number of sample points (cities)."**

**Comment 12:** Line 253 Figure 7: same units as for Fig 6 but now for country. Authors discuss correlations (agree, good in all cases) but changing axis extents/scales indicate USA comparable to Brazil, both greater than South Africa which is greater (brighter) than China and Australia both of which brighter than Italy. Confirm from other evidence? Or, if we should not indulge in quick visual speculations, use a uniform axis scale on all panels. Again, knowing n(umber) could prove very useful.

**Response:** Thanks for your suggestions. Originally, we focused on the accuracy evaluation of our product in this section, rather than the comparison among countries. So we used different axis scales in this figure. In this version, Figure 6 (original Figure 7) has been revised as you suggested. First, the number of sample pixels (n) were added in the new figure. Second, the axis scale on all panels were fixed as 0 to 150.

[Figure]

**Figure 6: The comparison between composited NPP-VIIRS NTL data and extended NPP-VIIRS-like NTL data in 2012 (unit: nano-Wcm⁻²sr⁻¹)** at the pixel level in six countries. Solid line denotes the 1:1 line **and N is the number of sample points (cities).**

**Comment 13:** Line 262 Figure 8: What city or cities. Same 400k cities used in Fig 6b? DN vs nanoWatt intensities so the axis scales have changed, correlation coefficients as before but now the solid red lines indicate data slopes not 1:1 lines. What do we make of the changing slopes. No mention from authors. We need n(umber) indication! 2004 tends to show outlying data points below the slope line, e.g. high intensities for low DN. 2006 and 2010 show opposite pattern: higher DN for lower intensities. Useful? Not useful? Perhaps slope lines have no value outside of specific chart but the apparent temporal variance of those lines 2000 to 2010 does not accord with Figure 10?

**Response:** Thank you for your questions. The same 40,000 cities used in Figure 5b (original Figure 6b) were also used here to validate our product in four years (2000, 2004, 2006, and 2010). For this validation, the DMSP-OLS RNTL with a pre-flight calibration was used as reference data. However, for the time-series analysis, an additional inter-calibration of DMSP-OLS RNTL data is still needed, due to the bias generated by sensor degradation and other sources (Feng-Chi et al., 2015). In this study, we evaluated our product using DMSP-OLS RNTL data in each year. The inconsistency in the DMSP-OLS RNTL data leads to the different slopes in Figure 7 (original Figure 8). The solid red lines indicate the fitting trends.

The corresponding statement in section 4.2 (around line 258-267) has been modified as "Fig. 7 shows that our extended NPP-VIIRS-like NTL data has a strong agreement with the DMSP-OLS RNTL data in the same 40000 cities in four years (2000, 2004, 2006, and 2010), which implies that the AE model is suitable for simulating NPP-VIIRS-like NTL data during the entire period. Before 2012, the composited NPP-VIIRS NTL data is not available for the validation, but the DMSP-OLS RNTL data is accessible in some separated years. To validate our results before 2012, we have to use the DMSP-OLS RNTL data as the reference data. The DMSP-OLS RNTL data was calibrated by using pre-flight sensor calibration and has no actual radiance

value. Thus, this validation was conducted at the city level, and the total NPP-VIIRS-like NTL intensity and DMSP-OLS RNTL intensity of each city were calculated and scattered (Fig. 7). In these four years, all the $R^2$ are higher than 0.75 and demonstrate our model does work for this entire time series. Note that the DMSP-OLS RNTL data still has inter-annual biases due to sensor degradation and other sources (Feng-Chi et al., 2015), resulting in the different slopes of trend lines in four years." Meanwhile, the Figure 7 (original Figure 8) has also been revised with adding the Slope value as "

[Figure]

**Figure 7: The comparison between the annual DMSP-OLS RNTL intensity (DN Value) and extended NPP-VIIRS-like NTL intensity (unit: nano-Wcm$^{-2}$sr$^{-1}$) at the city level in (a) 2000, (b) 2004, (c) 2006, and (d) 2010."**

Feng-Chi, H., Kimberly, B., Tilottama, G., Mikhail, Z., and Christopher, E.: DMSP-OLS Radiance Calibrated Nighttime Lights Time Series with Intercalibration, Remote Sensing, 7, 1855-1876, 10.3390/rs70201855, 2015.

**Comment 14:** Line 272 Figure 9: Authors justify use of median, but from where do these data come? Text says "based on 5000 random validation points." Random pixels? Apparently so, with axis scales at 0 to 150. All with data greater than 0, e.g. density of 'low' or greater? Density axis (again, population density?) refers in this case to city, not pixel? What do I not understand here?

**Response:** Thank you for your questions. The 5000 random points refer to validation pixels. As mentioned in Comment 11, density is the kernel density of the dots in scatter plot, rather than population density. Following the color rule of the density scale, the dots were colored according to their kernel density. The higher the density is, the warmer color the dot shows on the plots.

The description in section 4.2 (around line 269-270) has been revised as "Finally, a comparison between the composited NPP-VIIRS NTL data and the official annual NPP-VIIRS NTL data in 2015 was performed based on 5000 random validation pixels, and the result indicated the former is close to the latter (see Fig. 8)." In addition, the caption of Figure 8 (original Figure 9) has been modified as "Figure 8: A comparison of composited and official annual NPP-VIIRS NTL data via (a) histograms and (b) scatter plot with kernel density."

**Comment 15:** Line 278: the term "downtown" conveys neither geographic nor NTL density accuracy or specificity.

**Response:** Thank you for your comment. To clarify it, the sentence in section 4.3 (around line 282-284) has been modified as "In addition, our product can be used to explore the differences of NTL intensity among cities. For example, these three enlarged subplots clearly show that New York had higher NTL intensity than either of the other two cities."

**Comment 16:** Figure 10 very useful, lots of information. Text discussion of Fig 10 implies greatest NTL intensity change occurs globally over 2010 to 2015? Highest longitudinal peak might occur in Middle East (not = Europe), not mentioned in text?

**Response:** Figure 9(e) and (f), the original Figure 10, shows that the blue (2015) part is much larger than the yellow (2010) or green (2005) part. We can conclude that from 2010 to 2015, the world has a significant growth of NTL intensity. The global financial crisis of 2007-2008 has heavily hampered the global economic development until 2010. After that, the world returned to robust growth, which agrees with the previous research (Chen et al., 2019). In addition, Middle East shows the significant growth in the longitudinal direction.

The corresponding statement in section 4.3 (around line 290-295) has been modified as "In the eastern hemisphere region, there were three significant peaks in Europe, Middle East, and China (from west to east). The temporal changes of NTL intensity between 2000 and 2005 were generally slighter than that between 2005 to 2010. But from 2010 to 2015, the blue part (2015) in Fig. 9(e) and (f) was larger than the yellow part (2010), which implies that the NTL intensity strengthened almost all over the world. This result was highly associated with the global economic recovery after the global financial crisis of 2007-2008 (Chen et al., 2019a)."

Chen, W., Mrkaic, M., and Nabar, M. S.: The global economic recovery 10 years after the 2008 financial crisis, International Monetary Fund, 2019a.

**Comment 17:** Line 316, Figure 13: Potentially useful figure but quite confusing. ID units relative, not km? No explanation for the circled discrepancy in 13c? Values represent averages (means, medians?) along track at some spatial resolution. Or pixel by pixel? Y-axis scales change with every city; use a common scale to

allow city-by-city comparisons?

**Response:** Thank you for your suggestions. The NTL profiles were composed of each pixel's NTL intensity along with the solid or dashed line on the left map of Figure 12 (original Figure 13) and the ID indicates the pixel number from the start pixel. Note that the distance between two neighboring pixels was not fixed in this study, because the direction of the NTL profiles was not exactly parallel with the X-direction or Y-direction of pixels. Therefore, we did not use "km" as our x-axis in this figure.

As mentioned in the manuscript, the underestimated pixels were mostly in the urban periphery region, where the economy might have a rapid development, but the infrastructure construction could be lagged. This unbalanced development would lead to some extreme situations, such as a region only with a large building, but its surrounding was mostly bare area. In other words, the pixel where the building located could have a relative high NTL intensity, but its surrounding pixels mostly have a low NTL intensity. Under this extreme situation, these pixels would probabilistically be underestimated, because when we conducted a convolutional and deconvolutional operation, the surrounding pixels might lower the centric pixel's NTL intensity. Note that, such extreme situations are very rare, so the total accuracy of our product is still reliable. In the future research, the land cover/land use data could be of help to further improve our product's quality, because it can be used to distinguish the landscape patterns.

In the manuscript, the statement in section 5.1 (around line 322-326) has been modified as "This situation mostly appeared within the urban periphery region (e.g., ID: 80 – 90 in Fig. 12(c)) and could be caused by some extreme situations, such as dramatically unbalanced development, which leads to specific pixel with an abnormal high NTL intensity while the surrounding pixels have a relative low NTL intensity. Under such situation, the surrounding pixels might lower the centric pixel's NTL intensity, when the convolutional and deconvolutional operations were conducted." In section 5.3 (around line 372-373), we have added a sentence as "However, the land cover/land use data could be useful to mitigate the underestimations in Fig. 12(c), because these data can help distinguish the extreme development situation." Meanwhile, the figure has also been redrawn, including fixing the Y-axis scales from 0 to 200 for all profiles and adding the explanation of "ID" as "pixel number". The new figure is "

[Figure]

**Figure 12: Profiles of composited NPP-VIIRS NTL data and extended NPP-VIIRS-like NTL intensity (unit: nano-Wcm$^{-2}$sr$^{-1}$) across (a and b) Los Angeles, USA, (c and d) Shanghai, China, and (e and f) Cape Town, South Africa."**

**Comment 18:** Line 319: "nearby the junction point" what does this mean? End of simulated and start of composited?

**Response:** Yes. The "junction point" means the period when the simulated NPP-VIIRS-like NTL data ended and the composited NPP-VIIRS NTL data started. To clarify this term, we clarified the sentence in section 5.2 (around line 329-331) as "We compared the extended time-series of NPP-VIIRS-like NTL data with the time-series of census data and analyzed the range of NTL intensity change near the temporal joining point (that is, the final year of simulated NPP-VIIRS-like NTL data and the first year of the composited NPP-VIIRS NTL data, see the rectangular box in Fig. 14)."

**Comment 19:** Line 323: world bank population data. We need a version number, DOI, accurate references,

assurance of open access, … This data product should be referenced in improved Table 1, above?

**Response:** We have added the accurate reference and more detailed URL information in Table 1.

**Table 1: The list of used data in this study**

| Dataset | Source | Role |
|---|---|---|
| EANTLI | Calibrated DMSP-OLS NTL data[1]
EVI Data[2] | Input data in AE model (2000-2013) |
| Composited NPP-VIIRS NTL Data | Monthly NPP-VIIRS NTL data[3] | Reference data for validation (2012)
Label data in AE model (2013)
part of NPP-VIIRS-like NTL data (2013-2018) |
| DMSP-OLS RNTL Data[3] | F12-F15_20000103-20001229_rad_v4
F14_20040118-20041216_rad_v4
F16_20051128-20061224_rad_v4
F16_20100111-20101209_rad_v4 | Reference data for validation (2000, 2004, 2006, 2010) |
| Census Data[4] | Total Population (ID: SP.POP.TOTL) | Reference data for temporal consistency validation (2000-2018) |

[1.] Accessed from Li et al. (2020) in February 2020.

[2.] Accessed from MOD13A1 version 5 based on Google Earth Engine in May 2020.

[3.] Accessed from the Earth Observations Group (EOG) in Colorado School of Mines (https://payneinstitute. mines.edu/eog/nighttime-lights/) in May 2020.

[4.] Accessed from World Bank (2020) in May 2020.

**Comment 20:** Line 328 Figure 14. Ln:Ln plot gives little to no confidence. Linear trend of log-transformed data, is that even valid? Use an inset to show global data in separate non-log figure? Once reader gains confidence with global numbers, regional numbers will make more sense. Or separate into two panels, one for global and one for 7 countries. If this represents first use of population data, then what did density scales in Figs 6 and 9 represent? Some spatial clustering of high intensity pixels? What?

**Response:** Thank you for your comments. The ln:ln plot has been removed and two new separated panels have been added. One is scatter plot between population and total NTL intensity at the global scale and the other one is for seven countries. We also recalculated the $R^2$ of linear regression for globe and seven countries to confirm the extended time-series of NPP-VIIRS-like NTL data have a consistent temporal trend. The $R^2$ is 0.84 at global scale and ranges from 0.65 to 0.90 in seven countries, which implies that our NTL data have a good agreement with population without logarithmic transformation.

In the section 5.2 (around line 335-336), we have revised our statement as "The $R^2$ at the global scale is 0.84 (Fig. 13(a)) and the $R^2$ of seven selected countries (Fig. 13(b)) ranges from 0.65 (in United States and France) to 0.90 (in China)."

Meanwhile, the Figure 13 (original Figure 14) has been redrawn. It is worthy to note that because the

magnitude of NTL intensity in China and United States is much higher than that in other countries. We have to use two y-axes in Fig. 13(b). The right one is for China and United States, and the left one is for the other countries. The new figure is "

[Figure]

**Figure 13: Comparison of total population with total NTL intensity for (a) globe and (b) seven representative countries.**"

As mentioned above (Comment 11 and 14), the density scale in Fig. 5 and 8 (original Fig. 6 and 9) indicates the kernel density of dots in scatter plot.

**Comment 21:** Line 330: here a reader learns about "temporal junction point" and can use Figure 15 to understand. Need similar clarity at line 319.

**Response:** Thank you again. We have added a supplementary description to explain what "temporal junction point" is around line 329-331 (original line 319), as "We compared the extended time-series of NPP-VIIRS-like NTL data with the time-series of census data and analyzed the range of NTL intensity change near the temporal joining point (that is, the final year of simulated NPP-VIIRS-like NTL data and the first year of the composited NPP-VIIRS NTL data, see the rectangular box in Fig. 14)."

**Comment 22:** Lines 334 to 336: Do the time-dependent changes discussed here, strong from 2010 to 2014 but diminished from 2014 to 2018, accord with what the authors concluded earlier, e.g. about changes globally and regionally from 2010 to 2015 as in Fig 10?

**Response:** Thank you for your questions. In Figure 9 (original Figure 10), we focused on the dynamics of NTL intensity during this period, and we found the NTL intensity showed a strong increase from 2010 to 2014 and became stable from 2014 to 2018. In line 340-342 (original line 334-336), we explored NTL intensity fluctuation nearby the temporal junction point for evaluating the temporal consistency of combined our product. Overall, these two statements are consistent.

**Comment 23:** Line 346 Figure 15. Same six countries used in Fig 7? Why these six? Y-axis scale varies in both minimum and maximum in almost every plot. Ln values? Not helpful! Again, one scale for global and a second uniform scale for all countries? These apparent offsets per country with time explain the different slopes in Figure 7?

**Response:** First, we selected one representative country in each continent. Second, we improved Figure 14 (original Figure 15) as you suggested. The log-transformed total NTL intensity and lit pixel were replaced by the original one and the scales for all plots within the six countries were fixed (Figure R1). However, because the magnitudes highly varied among countries, using the uniform scales for all countries, the lines

in Brazil, Italy, South Africa, and Australia have been compressed in a small range.

[Figure]

**Figure R1: Temporal trend of extended time-series (2000-2018) NPP-VIIRS-like NTL data for the total NTL intensity and lit pixels at (a) the global scale and within six selected countries: (b) United States, (c) Brazil, (d) Italy, (e) South Africa, (f) China, and (g) Australia.**

Therefore, we decided to set three scales for these six countries in this version to show the fluctuation in the four countries mentioned above. Specifically, United States and China have the same scale in Fig. 14(b) and (c), Italy and Brazil have the same scale in Fig. 14(d) and (e), and the rest two countries have the same scale in Fig. 14(f) and (g).

[Figure]

**Figure 14:** Temporal trend of extended time-series (2000-2018) NPP-VIIRS-like NTL data for the total NTL intensity and lit pixels at (a) the global scale and within six selected countries: (b) United States, (c) China, (d) Italy, (e) Brazil, (f) South Africa, and (g) Australia.

Third, in Figure 14 (original Figure 15), the blue line and red line indicate the total NTL intensity and the number of lit pixels from our extended NPP-VIIRS-like NTL (2000-2018), respectively. In Figure 6 (original Figure 7), the slope refers to the relationship between our extended NPP-VIIRS-like NTL intensity and the composited NPP-VIIRS NTL intensity of sample pixels in 2012. We thought these two figures cannot be compared directly. The offset between two lines in Figure 14 does not imply the slope in Figure 6.

**Comment 24:** Small language issues throughout (e.g. line 49 comparable not comparability, "worthy to note" rather than noteworthy in several places, etc.); one hopes / assumes Copernicus pre-publication efforts will clean these up?

**Response:** Thanks for your suggestion. We improved the language of the manuscript and all the revision were marked as red. For example, "comparability" in line 50 has been modified as "comparable", "worthy noted" in line 121 has been modified as "worth noting", and "an kernel density-based" in line 70 has been modified as "a kernel density-based".

**Response to Review ESSD-2020-201-RC2**

This paper introduces an extended time-series (2000-2018) of NPP-VIIRS-like NTL data through a cross-sensor calibration from DMSP-OLS NTL data (2000-2012) and a composition of monthly NPP-VIIRS NTL data (2013-2018). The authors claim that compared with the annual composited NPP-VIIRS 20 NTL data in 2012, their product of extended NPP-VIIRS-like NTL data shows a good consistency at the pixel and city levels with R2 of 0.87 and 0.95, respectively. The paper is well organized and written clearly, and the data set should be of interest to users who use the NTL data. Therefore, the paper should be accepted for publication after the follow issue is addressed.

**Response:** Thank you very much for your suggestions and comments. The manuscript has been revised carefully according to your comments. Please see our response and revision below.

**Comment 1:** It is strongly recommended that a short description of the uniqueness of the method used should be included in the abstract. The current abstract only tells the reader that there is a new product and it is better, but it fails to say what it is and how it is produced. For example, if the uniqueness is that they relied on the vegetation index adjusted NTL to perform the simulation, it should say so up front in the abstract, so that readers can get the main point without having to read the entire paper. Other than that, the paper is well written.

**Response:** Thanks for your suggestion. In this study, we developed a new cross-sensor calibration method using image enhancement. This method includes four key steps. In the Step 1 we adjusted the DMSP-OLS NTL data using a vegetation index, named as the enhanced vegetation index-adjusted NTL index (EANTLI). In the Step 2, the EANTLI and NPP-VIIRS NTL data (2012-2013) were used as the input and label data of auto-encoder (AE) model. By training the AE model, a model of NPP-VIIRS-like NTL data simulation was built. In the Step 3, driven by the EANTLI from 2000 to 2012 in the trained AE model, the NPP-VIIRS-like NTL data was built. Finally, by merging with the composited annual NPP-VIIRS NTL data (2013-2018), an extended time-series (2000-2018) of NPP-VIIRS-like NTL data were built.

We have added a short statement of our methodology in the abstract, as "However, the difference in their spatial resolutions and sensor design requires a cross-sensor calibration of these two datasets for analyzing a long-term urbanization process. Different from the traditional cross-sensor calibration of NTL data by converting NPP-VIIRS to DMSP-OLS-like NTL data, this study built an extended time-series (2000-2018) of NPP-VIIRS-like NTL data through a new cross-sensor calibration from DMSP-OLS NTL data (2000-2012) and a composition of monthly NPP-VIIRS NTL data (2013-2018). The proposed cross-sensor calibration is unique due to the image enhancement by using a vegetation index and an auto-encoder model."